# A CROSS-SPECIES NEURAL FOUNDATION MODEL FOR END-TO-END SPEECH DECODING

**Yizi Zhang**[*,1,†]**, Linyang He**[*,1,†]**, Chaofei Fan**[2]**, Tingkai Liu**[3]**, Han Yu**[1]**, Trung Le**[4]**, Jingyuan Li**[5]**,
Scott Linderman**[2]**, Lea Duncker**[1]**, Francis R Willett**[2]**, Nima Mesgarani**[1]**, Liam Paninski**[1]
[1]Columbia University, New York, NY, USA
[2]Stanford University, Palo Alto, CA, USA
[3]Microsoft, New York, NY, USA
[4]University of Washington, Seattle, WA, USA
[5]Amazon, Seattle, WA, USA
*_Equal contribution_
[†]{yz4123, lh3288}@columbia.edu

## ABSTRACT

Speech brain–computer interfaces (BCIs) aim to restore communication for people with paralysis by translating neural activity into text. Most systems use cascaded frameworks that decode phonemes before assembling sentences with an n-gram language model (LM), preventing joint optimization of all stages simultaneously. Here, we introduce an end-to-end **BraIn-to-Text (BIT)** framework that translates neural activity into coherent sentences using a single differentiable neural network. Central to our approach is a cross-task, cross-species pretrained neural encoder, whose representations transfer to both attempted and imagined speech. In a cascaded setting with an n-gram LM, the pretrained encoder establishes a new state-of-the-art (SOTA) on the Brain-to-Text '24 and '25 benchmarks. Integrated end-to-end with audio large language models (LLMs) and trained with contrastive learning for cross-modal alignment, BIT reduces the word error rate (WER) of the prior end-to-end method from 24.69% to 10.22%. Notably, we find that small-scale audio-LLMs markedly improve end-to-end decoding. Beyond record-setting performance, BIT aligns attempted and imagined speech embeddings to enable cross-task generalization. Altogether, our approach advances the integration of large, diverse neural datasets, paving the way for an end-to-end decoding framework that supports seamless, differentiable optimization.

## 1 INTRODUCTION

Speech neuroprosthetics have experienced a recent revolution, enabling the decoding of attempted and imagined speech and restoring communication for those who have lost the ability to speak (Akbari et al., 2019; Makin et al., 2020; Moses et al., 2021; Proix et al., 2022; Willett et al., 2023; Fan et al., 2023; Metzger et al., 2023; Card et al., 2024; Littlejohn et al., 2025; Kunz et al., 2025). Despite this success, most existing systems rely on cascaded frameworks that cannot easily be optimized end-to-end. In such models, recurrent neural networks (RNNs) map neural activity to phonemes that are then assembled into sentences with an n-gram LM (Willett et al., 2023). While efficient for small datasets, separate optimization of each component can result in a dissociation between the performance of the RNN and the system as a whole. For instance, lower phoneme error rates (PER) from the RNN do not always translate to lower word error rates (WER) when decoding with the n-gram model (Willett et al., 2024). These limitations underscore the need for a fully end-to-end optimized speech decoding framework.

Recently, Feng et al. (2024) connect RNNs with LLMs to translate neural activity into sentences in an end-to-end manner. While promising, this approach does not explore modern architectures, such as transformers, which may better capture complex neural representations and enhance decoding performance. In contrast, Feghhi et al. (2025) incorporate transformers to decode phonemes and apply time masking to mitigate overfitting, achieving higher performance, but their study does not employ end-to-end training with LLMs or leverage large-scale pretraining. Since transformers typi-

cally require large datasets to reach their full potential, we hypothesize that combining transformer-based neural encoders with large-scale pretraining could further improve decoding performance and provide stable representations to effectively guide LLMs (Liu et al., 2023b).

Here, we present an end-to-end **B**ra**I**n-to-**T**ext neural interface that we call **BIT**, which translates neural activity directly into coherent sentences. Our approach introduces several key innovations: a transformer neural encoder pretrained with self-supervised masked modeling on 367 hours of Utah array recordings from humans and monkeys across speech and arm-related motor tasks; and, after fine-tuning, a flexible decoder that supports either a cascaded framework, where decoded phonemes are assembled into sentences with an n-gram LM, or an end-to-end framework, where an audio-LLM decoder directly generates sentences with contrastive alignment between neural and text embeddings. Analogous to LLAVA (Liu et al., 2023b), which augments a pretrained LLM with an image encoder as its eyes for visual perception, BIT equips the LLM with a brain to interpret neural activity. Through transformer-based pretraining and end-to-end LLM integration, BIT overcomes the limitations of RNN-only encoders and small-scale datasets, yielding neural representations that generalize across tasks and align speech-related neural activity with the language structure of LLMs.

We evaluate our approach on the Brain-to-Text Benchmark '24 and '25 datasets (Willett et al., 2023; Card et al., 2024), which consist of Utah array recordings from two human participants attempting to speak sentences cued on a computer monitor. First, we benchmark the pretrained encoder on attempted speech decoding, demonstrating that it outperforms both RNNs and transformers trained from scratch in the cascaded framework. Next, we evaluate LLM decoders, showing that small-scale audio-LLMs substantially improve end-to-end decoding performance, surpassing the text-based LLM used in prior work (Feng et al., 2024). These gains are even more pronounced when transferring to imagined speech (Kunz et al., 2025), a low-data task with far fewer labeled examples. Interpretability analysis reveals that BIT's neural embeddings for attempted and imagined speech align through shared semantic structure. These findings establish BIT as a new paradigm for integrating intracortical BCIs with LLMs, enabling neural activity to guide sentence generation without relying on intermediate phoneme or character outputs. Overall, BIT represents a step toward end-to-end speech BCIs that translate neural activity directly into coherent sentences.

## 2 RELATED WORK

**Speech brain-computer interfaces.** Early speech BCIs transform neural signals from auditory and motor cortices into intelligible outputs, enabling real-time and high-performance brain-to-text communication (Bouchard et al., 2013; Akbari et al., 2019; Makin et al., 2020; Moses et al., 2021; Willett et al., 2023; Fan et al., 2023; Card et al., 2024). These cascaded systems typically map activity to phonemes or characters before language model composition. Recent advances introduce transformers for long-range dependencies, streaming implementations, and end-to-end decoding with LLMs (Feghhi et al., 2025; Littlejohn et al., 2025; Feng et al., 2024). Finally, inner speech decoding raises the possibility of communication without attempting to move (Kunz et al., 2025). However, most prior work has relied on cascaded systems that cannot be trained end-to-end, while the previous end-to-end decoder, in turn, has relied on RNN encoders without pretraining, not taking advantage of modern model architectures and training approaches. Our work closes this gap with a new end-to-end framework powered by cross-species, cross-task transformer-based pretraining.

**Large-scale models for neural analysis.** Increasing attention has been directed toward large-scale pretraining to learn generalizable neural representations across modalities including fMRI, calcium imaging, spiking activity, EEG, and EMG (Scotti et al., 2024; Thomas et al., 2022; Lurz et al., 2020; Azabou et al., 2023; 2024; Ryoo et al., 2025; Ye et al., 2023; 2025; Zhang et al., 2024; 2025; Cui et al., 2024; Kaifosh & Reardon, 2025). These emerging "foundation models" improve performance while reducing reliance on task-specific data, and have shown success in behavior decoding, motor imagery, visual reconstruction, region and cell-type classification, mental state decoding, and neural activity prediction (Thomas et al., 2022; Scotti et al., 2024; Cui et al., 2024; Azabou et al., 2024; Yu et al., 2025; Wang et al., 2025). In spiking data, POSSM (Ryoo et al., 2025) has demonstrated cross-species transfer from monkey to human imagined handwriting. Building on this progress, we explore cross-species, cross-task self-supervised pretraining for decoding human speech.

**Modality alignment in multimodal language models.** Recent advances in visual language models (VLMs) illustrate how large pretrained models can integrate disparate modalities. Early models such as CoCa (Yu et al., 2022) rely on cross-attention projector for aligning billions of image–text pairs, achieving strong image-captioning performance but at high computational cost. Li et al. (2023)

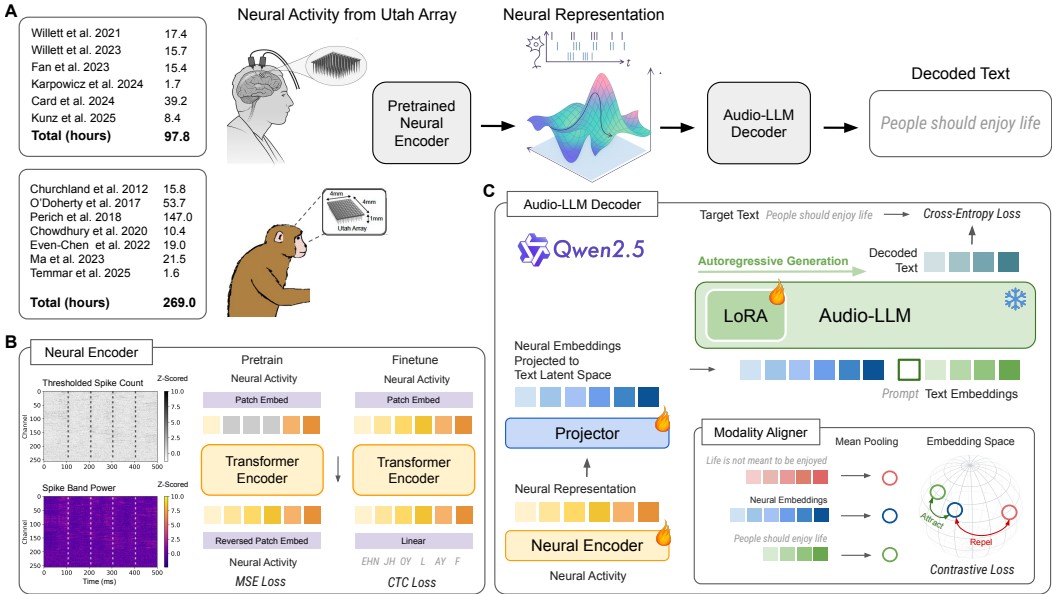

Figure 1: *Schematic illustration of BIT.* **(A)** BIT is an end-to-end speech decoding framework that translates neural activity directly into text by combining a cross-task, cross-species pretrained neural encoder with an audio-LLM decoder. The data are separately obtained and preprocessed from each study. (Appendix A). **(B)** The neural encoder is a transformer that embeds 20 ms bins of thresholded spikes and spike-band power into multi-bin time patches. It is pretrained using SSL with time-patch masking, reconstructing patch tokens via subject-specific linear read-in and read-out layers with an MSE loss. After pretraining, the masking module is removed, and the encoder is fine-tuned for phoneme decoding using a linear classifier with CTC loss. **(C)** The neural encoder outputs are mapped to the text embedding space of an audio-LLM via a shallow MLP projector. A modality aligner trained with contrastive learning projects mean-pooled neural and text embeddings into a shared latent space for modality alignment. To guide decoding, we insert a prompt between neural and text embedding tokens: *"decode the above neural activity into an English sentence:".* During finetuning, we update the neural encoder, projector, and apply LoRA to the linear layers in the audio-LLM's attention and feed-forward blocks, while keeping other parameters frozen.

introduce a more efficient design by freezing pretrained visual and language backbones and training a compact Q-former to align modalities. More recently, instruction-tuned LLMs such as LLaVA (Liu et al., 2023a) replace query-based intermediates with direct projection layers mapping visual features into the LLM embedding space. This progression, from cross-attention to Q-formers to simple projections, demonstrates that as LLMs become more capable, modality alignment becomes less compute-intensive. Inspired by these trends, we investigate whether LLMs can similarly generalize to speech-related neural signals.

## 3 METHODS

In this section, we present an end-to-end speech BCI consisting of a transformer-based neural encoder and an audio-LLM decoder for speech translation (Figure 1). The encoder and decoder are connected by a shallow MLP, trained with contrastive learning for modality alignment. The transformer encoder is first pretrained with self-supervised masked modeling and then fine-tuned for phoneme decoding. Finally, this phoneme-aware encoder is fine-tuned to translate coherent sentences directly from neural activity using the audio-LLM.

### 3.1 DATASET

The Brain-to-Text Benchmark '24 and '25 datasets (Willett et al., 2023; Card et al., 2024) are used for attempted speech decoding. The '24 dataset includes 12,100 sentences from participant T12 across 25 sessions over four months, with 128-electrode Utah array recordings from ventral motor cortex. The '25 dataset contains 10,948 sentences from participant T15 across 45 sessions over 20 months, recorded with 256 electrodes in ventral motor cortex. For imagined speech, we use the inner speech dataset (Kunz et al., 2025), comprising 500 and 712 sentences for T12 and T15, recorded with 128 or 256 electrodes while participants internally spoke cued sentences. For all

datasets, neural features include thresholded spike counts and spiking-band power (SBP) (Nason et al., 2020), binned in 20 ms windows and normalized across days (z-scored) to address Utah array non-stationarity (Karpowicz et al.). For pretraining, we use $\sim 98$ hours of human Utah array recordings (including the above speech BCI datasets) and $\sim 269$ hours of monkey recordings during motor tasks (Willett et al., 2021; 2023; Card et al., 2024; Fan et al., 2023; Karpowicz et al., 2024; Kunz et al., 2025; Churchland et al., 2012; Chowdhury et al., 2020; Even-Chen et al., 2019; Ma et al., 2023; Perich et al., 2018). Since SBP is unavailable in some pretraining datasets, we use only thresholded spikes, normalized across days. See Appendix A for details.

## 3.2 ARCHITECTURE

**Transformer neural encoder.** We employ a transformer to learn latent representations from neural activity. The original data has shape $(T, C)$, where $T$ is the number of time bins and $C$ is the number of electrodes. Following Feghhi et al. (2025), we group every $T_{\text{patch}}$ time bins into a patch, yielding data of shape $(T/T_{\text{patch}}, C \times T_{\text{patch}})$. These patches are passed through a patch embedding module (LayerNorm, Linear, LayerNorm) to produce tokens for the transformer encoder with multi-headed attention. Using time patches rather than individual 20 ms bins aligns the finer temporal resolution of neural recordings with the slower timescale of speech production (30–60 words per minute). In addition, time patch tokens shorten the context length, minimizing information redundancy when feeding neural embeddings to LLMs for short sentence generation.

Each transformer block includes a self-attention layer followed by a feed-forward network (Appendix Q). Relative positional embeddings (RoPE) (Su et al., 2024) encode temporal information, and a bidirectional attention mask allows each time patch to attend to all others. For self-supervised pretraining, neural activity from each subject is converted into transformer inputs via the patch embedding module, and the transformer outputs are reconstructed to the original neural data through the reversed patch embedding module, which projects latent tokens back to the input dimensionality. For phoneme decoding, transformer outputs are passed through a linear layer to generate logits over phonemes, the blank token, and the silence token.

**LLM decoder.** For text-LLMs, encoder outputs are mapped into the text embedding space via a shallow MLP projector (Linear, ReLU, Linear). Text is tokenized and embedded in the same space. A text prompt is inserted between neural and text embeddings to guide decoding, with design varying depending on whether encoder outputs are treated as distinct modalities (Figure 3B). During training, the model receives neural embeddings, text embeddings from prompt and target sentence, and generates text tokens via next-token prediction. At inference, the model generates decoded text tokens autoregressively using only the neural embeddings and prompt. For audio-LLMs, neural encoder outputs can be treated as a distinct ***neural modality*** and mapped into the text space, following the same procedure as for text-based LLMs. Since the encoder is pretrained on phoneme decoding, the resulting embeddings encode phonetic information. Even without audio tokens, the model can leverage speech knowledge acquired during LLM pretraining. When treated as an ***audio modality***, neural encoder outputs first pass through the MLP projector and are then mapped into the audio embedding space via the multimodal projector originally used for the audio encoder (Figure 3A). In this case, neural activity can be treated as audio since it resembles speech waveforms.

We apply low-rank adaptation (LoRA) (Hu et al., 2022) to a subset of LLM parameters, including attention projections and feed-forward layers, to enable fine-tuning with few trainable parameters. For audio-based LLMs, LoRA is also applied to the multimodal projector, mapping neural embeddings into the audio embedding space to align with the model's acoustic representations.

**Cross-modal alignment.** After projecting encoder outputs into the text embedding space, a modality aligner projects mean-pooled neural and text embedding tokens into a shared latent space using separate linear layers. The resulting L2-normalized vectors are optimized with a contrastive loss to increase similarity between embeddings of matching sentences (Figure 1C). See Section 3.3 and Appendix I for details.

## 3.3 TRAINING OBJECTIVES

**Self-supervised pretraining using masked modeling.** For pretraining, we utilize a temporal masking strategy inspired by masked autoencoders (He et al., 2022). Neural activity is tokenized into temporal patches, and a subset is randomly replaced with a learnable mask token. Masked patches can form contiguous spans of variable length, up to a predefined maximum timespan, while main-

taining a consistent overall masking ratio (Appendix Q). The model is then trained to reconstruct the original neural activity from these partially masked sequences, encouraging the learning of rich contextual representations without external supervision. We use Mean Squared Error (MSE) as the reconstruction loss, since both thresholded spike counts and SBP are normalized. This masking strategy reduces overfitting through data augmentation and mitigates non-stationarity from probe drift (Karpowicz et al.; 2024). Additionally, pretraining is limited to human and primate Utah array data, allowing the model to learn stable probe representations that are robust to variations in electrode placement, subjects, and behavioral tasks.

**Finetuning for phoneme-level decoding.** After pretraining, the neural encoder is finetuned to predict phoneme sequences from neural activity using a Connectionist Temporal Classification (CTC) loss (Graves et al., 2006). The encoder transforms neural activity into latent embeddings, which are projected onto phoneme classes plus a blank token to produce phoneme logits. During finetuning, the time masking module is removed, since pretraining with extensive data augmentations has already mitigated overfitting. In the end-to-end model, phoneme logits are not fed to the audio-LLM decoder; rather, they serve as intermediate targets that encode phonetic information into neural representations to guide LLM fine-tuning, even when the goal is sentence prediction (Appendix C).

**Finetuning for sentence-level decoding.** In the final stage, the neural encoder feeds neural representations into the audio-LLM decoder, which is fine-tuned to predict the next token using cross-entropy loss against the ground-truth sentence. This objective aligns neural activity with the language structure in LLMs, enabling coherent text generation. To reinforce modality alignment, we utilize a contrastive learning objective. The modality aligner in Section 3.2 pools neural and text embedding tokens into single "modality tokens" and projects them into a shared latent space. Positive pairs are formed by neural and text embeddings from the same trial corresponding to the same sentence, while all other examples in the batch serve as negative pairs. This encourages the model to pull matching embeddings closer and push non-matching embeddings apart, allowing the model to learn representations that align with the semantic structure in the text embeddings. The total loss is the sum of the cross-entropy and contrastive losses (Appendix J).

## 4 EVALUATION

### 4.1 TASKS

We evaluate our approach on two speech decoding tasks. For ***attempted speech***, neural activity is decoded as the participant physically tries to articulate words, although no intelligible audio output is produced due to paralysis (Willett et al., 2023; Card et al., 2024). For ***imagined speech***, neural activity is decoded as the participant silently imagines speaking without attempting to engage the speech muscles (Kunz et al., 2025). Performance is measured by word error rate (WER) averaged across sentences, with lower WER indicating more accurate decoding.

### 4.2 BASELINES

We benchmark our method against baselines using two decoding strategies:

**Cascaded decoder.** We first decode phonemes from neural activity using a neural encoder. Phoneme logits are passed into a 5-gram LM to generate sentence hypotheses, re-scored with OPT (Zhang et al., 2022) (Appendix D). Although not fully differentiable, this approach leads to the lowest WER for Utah-array speech BCIs (Willett et al., 2024) and is used to benchmark baseline encoders:

1. ***RNN***: Baseline RNN from prior speech BCIs (Willett et al., 2023; Card et al., 2024).
2. ***BIT-TFS***: Transformer trained from scratch for phoneme decoding.
3. ***BIT-Human***: Transformer pretrained on human data, fine-tuned for phoneme decoding.
4. ***BIT-All***: Transformer pretrained on human + monkey data, fine-tuned to decode phonemes.
5. ***BIT-Cross-Task-Only***: Transformer is trained to decode attempted speech from the same subject, and then fine-tuned for decoding imagined speech to test cross-task transfer.

**End-to-end decoder.** The neural encoder trained for phoneme decoding is paired with a pretrained LLM, fine-tuned to translate neural activity directly into sentences by feeding neural representations, rather than phoneme logits, into the LLM. We benchmark the same neural encoders (*RNN, BIT-TFS, BIT-Human, BIT-All, BIT-Cross-Task-Only*) with various text- and audio-based LLMs (Section 5.2) using nucleus sampling (Holtzman et al., 2020) for sentence generation at inference time (Appendix F). Although end-to-end methods (Feng et al., 2024) have not yet surpassed the cascaded decoder, our approach aims to narrow this gap.

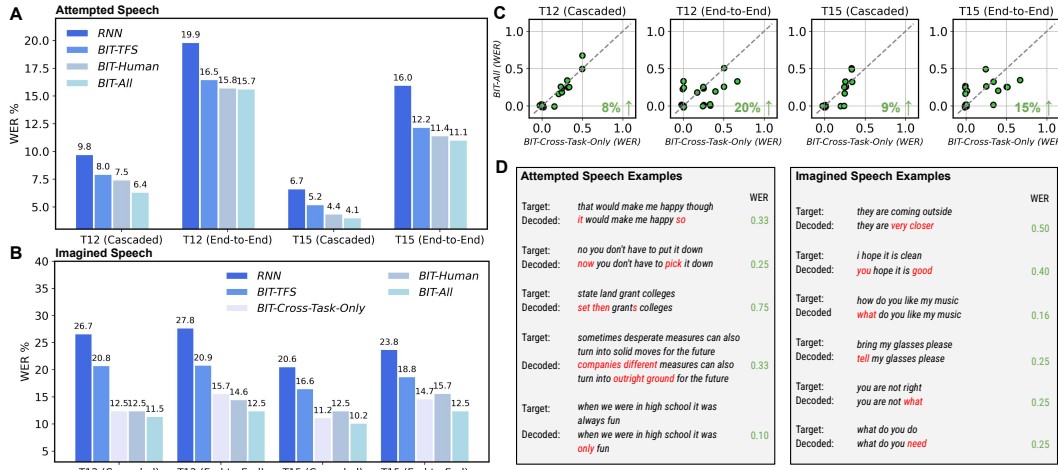

Figure 2: *Benchmarking BIT versus baselines in attempted and imagined speech decoding.* **(A)** For attempted speech, the pretrained encoder (*BIT-Human*, *BIT-All*) outperforms RNN and *BIT-TFS* using both cascaded and end-to-end approaches. Bar plots show mean WER across competition holdout sentences. **(B)** For imagined speech (50-word vocabulary), *BIT-All* outperforms all other baselines in both cascaded and end-to-end settings. Bar plots show mean WER across partitioned test sentences. SSL pretraining provides greater benefits for imagined speech than for attempted speech, since imagined speech has far fewer labeled examples. **(C)** Scatterplots compare *BIT-All* vs. *BIT-Cross-Task-Only* on imagined speech decoding, with each dot representing a test sentence and the green value showing relative improvement. Results show that SSL pretraining (cross-subject, unlabeled) yields larger transfer gains than SL pretraining (within-subject, cross-task) after fine-tuning. **(D)** Example decoded sentences from *BIT-All* using end-to-end approach. The imagined speech task has a smaller vocabulary (50 words) than attempted speech.

## 5 EXPERIMENTS AND RESULTS

### 5.1 DECODING ATTEMPTED AND IMAGINED SPEECH

To benchmark attempted and imagined speech decoding, models are trained and evaluated separately for each participant (T12 and T15). For attempted speech, we train on the Brain-to-Text '24 and '25 benchmark training datasets, use the provided test set as the validation set for model selection, and report performance on the benchmark's holdout set (ground truth is unknown, but we obtain scores via submission to the competition). For imagined speech, we use the training data provided by the original study, and split the evaluation set in half to create validation and test sets. WERs are computed on the holdout or test trials for each participant. For ablation studies, we report performance on the validation set, as it is used to select the final model for benchmark submission. See Appendix A for pretraining data splits.

For attempted speech, we compare *BIT-All*, *BIT-Human*, *BIT-TFS*, and the RNN baseline. For imagined speech, we also include *BIT-Cross-Task-Only* to evaluate within-subject, cross-task transfer. For both RNN and trans-

Table 1: *We benchmark BIT on the Brain-to-Text '24 hold-out set (1200 sentences), comparing it to other speech BCIs within the same decoding framework (cascaded or end-to-end) for fairness. Background colors indicate comparable methods. See the competition leaderboard for more details.*

| Competition Entry | WER |
|---|---|
| Feng et al. (2024) | 24.69% |
| BIT End-to-End | 15.67% |
| **BIT End-to-End + Ensemble** | **10.22%** |
| RNN (Baseline) | 9.76% |
| Feghhi et al. (2025) | 7.98% |
| **BIT Cascaded** | **6.35%** |
| Li et al. (2024) + Ensemble | 5.77% |
| Feghhi et al. (2025) + Ensemble | 5.68% |
| **BIT Cascaded + Ensemble** | **5.10%** |

former architectures, we perform hyperparameter tuning by initializing 30 models with optimizer hyperparameters randomly sampled from predefined ranges. We select the configuration with the best validation performance. See Appendix Q-R for details. In our experiments, *BIT-TFS* outperforms the RNN on attempted speech decoding (Figure 2A); since the RNN does not use time masking, this comparison reflects both architectural and training objective differences. Pretraining at scale (*BIT-Human*, *BIT-All*) further improves performance over training the encoder from scratch. For imagined speech (50-word vocabulary), *BIT-Human* and *BIT-All* outperform *BIT-TFS* by 39-

45% in WER, showing that SSL pretraining is especially beneficial in low-data settings. Moreover, *BIT-All* exceeds *BIT-Cross-Task-Only* (Figure 2B–C), indicating that cross-subject, label-free SSL pretraining yields larger transfer gains than same-subject, cross-task supervised pretraining. Although BIT surpasses the prior end-to-end baseline (Feng et al., 2024) combining an RNN encoder with a text-based LLM decoder, a gap in WER remains between cascaded and end-to-end approaches. Our method substantially narrows this gap, bringing end-to-end performance closer to that of the cascaded decoder. Figure 2D shows example sentences decoded by *BIT-All* in the end-to-end framework.

To benchmark against other methods, we participate in the Brain-to-Text '24 and '25 competitions, where WER is measured on a holdout set with ground-truth labels hidden from participants. For fair comparisons, we compare methods in the same decoding framework (cascaded or end-to-end); see Appendix H for details. In Tables 1 and 2, *BIT Cascaded* (5-gram LM) and *BIT End-to-End* (Aero1-Audio 1.5B) couple our pretrained encoder (*BIT-All*) with different decoders. *BIT Cascaded* achieves a SOTA WER of 6.35% for non-ensembled models, surpassing the previous best of 7.98% (Feghhi et al., 2025). With a fine-tuned LLM that merges multiple decoded sentences from an ensemble of models trained with different seeds (Appendix G), *BIT Cascaded + Ensemble* further reduces WER to 5.10%, outperforming the prior 5.68% (Feghhi et al., 2025) and ranking first. *BIT End-to-End + Ensemble* reduces WER from 24.69% to 10.22% compared to the previous end-to-end entry (Feng et al., 2024). On the '25 public leaderboard, *BIT Cascaded + Ensemble* leads with 1.76% WER, while *BIT End-to-End + Ensemble* achieves 7.76%.

Table 2: *We benchmark BIT on the Brain-to-Text '25 hold-out set (1450 sentences), comparing it to other speech BCIs within the same decoding framework (cascaded or end-to-end) for fairness. Background colors indicate comparable methods. See the competition leaderboard for more details.*

| Competition Entry | WER |
|---|---|
| BIT End-to-End | 11.06% |
| **BIT End-to-End + Ensemble** | **7.76%** |
| RNN (Baseline) | 6.67% |
| **BIT Cascaded** | **4.06%** |
| RNN-TTA + Pseudo-Ensemble | 4.42% |
| RNN + Ensemble | 3.09% |
| **BIT Cascaded + Ensemble** | **1.76%** |

## 5.2 ABLATING LARGE LANGUAGE MODEL DECODER

We conduct an ablation study to quantify the effects of LLM decoder type and model design on attempted speech decoding performance. Specifically, we compare text-based and audio-based LLMs, and test different prompt designs by treating neural embeddings as either audio or neural modalities (Section 3.2 and Figure 3B). Finally, we evaluate how contrastive learning for cross-modal alignment influences decoding performance. We focus on models with $1 \sim 7B$ parameters, which strike a balance between expressive capacity and adaptability for neural decoding with scarce labeled data. For text-based LLMs, we use Qwen2.5-1.5B, Qwen2.5-7B (Qwen et al., 2025), Qwen3-0.6B, and Qwen3-1.7B (Yang et al., 2025). For audio-LLMs, we use Aero1-Audio 1.5B (Li et al., 2025), an audio-extended version of Qwen2.5-1.5B, and Qwen2-Audio-7B (Chu et al., 2024), a larger model capturing richer acoustic representations. Comparing these audio-extended models with their text-only counterparts allows us to isolate the impact of audio processing from general language modeling capacity.

Our experiments (Figure 3C–D) show that, for models of comparable size, audio-based LLMs outperform text-based LLMs in decoding, with Aero1-Audio 1.5B (Li et al., 2025), the audio-extended variant of Qwen2.5 1.5B, consistently achieving the best results. This suggests that pretraining LLMs on audio tasks provides inductive biases closer to the neural decoding problem, enabling a shallow MLP to align modality more effectively. Treating neural encoder outputs as a neural modality performs slightly better than treating them as audio, indicating that while neural embeddings need not be interpreted as audio, LLMs still benefit from speech knowledge. We also find that smaller LLMs generally achieve lower WERs than larger LLMs when trained with limited labeled examples, since speech BCI tasks require English translation rather than advanced reasoning, which is typically seen in models above 7B parameters. Finally, using contrastive learning to align neural and text embeddings further improves performance.

## 5.3 ALIGNING NEURAL EMBEDDINGS FOR CROSS-TASK GENERALIZATION

Although BIT is trained for speech decoding, we want to understand what semantic content is encoded in its latent embeddings. To this end, we conduct sentence-level representational similarity

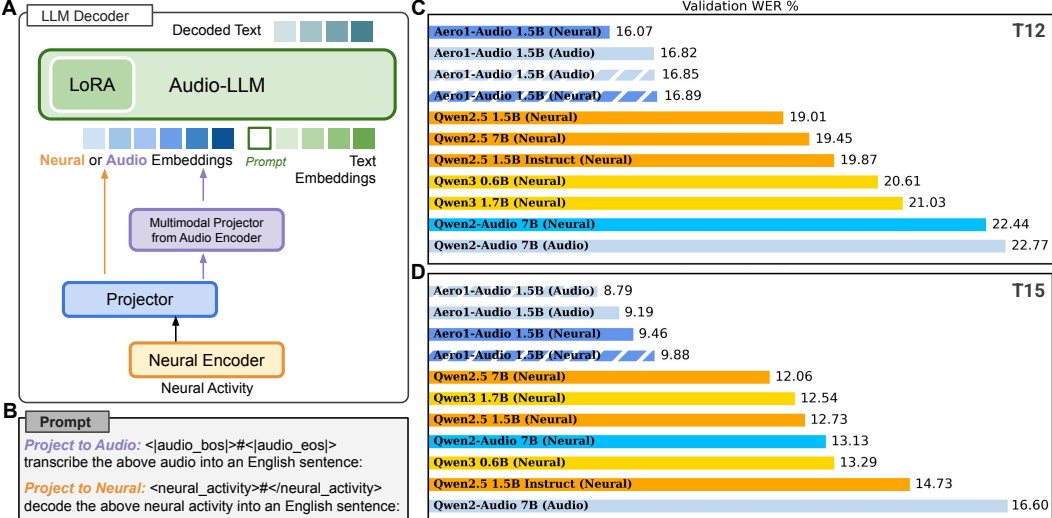

Figure 3: *LLM decoder ablation across modality, model size, prompt design, and contrastive learning usage.* **(A)** For audio-LLMs, neural activity can be treated as either a neural or an audio modality. For neural modality, encoder outputs are projected directly into the text embedding space via an MLP projector. For audio modality, neural encoder outputs pass through the MLP projector followed by a multimodal projector used for the audio encoder. **(B)** Different prompts are used depending on whether neural encoder outputs are treated as neural or audio modality, with the encoder outputs inserted at the placeholder "#". **(C-D)** Bar plots show the mean WER across validation sentences for different LLM models, modality treatments, prompt designs, and contrastive learning usage. Here, we report validation WER, as it is used to select the final LLM decoder for benchmark submission. Colors distinguish text- (yellow) and audio-LLMs (blue), with transparency indicating whether neural activity is treated as *Neural* or *Audio* modality, while diagonal hatching denotes that contrastive learning is not used.

analysis (RSA) (Kriegeskorte et al., 2008) to compare the representational geometry of neural embeddings against that of LLMs. If neural embeddings preserve relational structures similar to LLMs, they may provide more informative features for decoding. Importantly, all neural encoders in the RSA analysis are taken *prior to sentence-level fine-tuning* to ensure that we compare the intrinsic representational geometry rather than the effect of supervised adaptation (details in Appendix L.1). Figure 4A shows the RSA scores between neural and audio-LLM text embeddings. Pretrained encoder outputs are more similar to audio-LLM representations than those of RNN and *BIT-TFS*, indicating that large-scale pretraining facilitates linguistically structured representations.

We also want to understand which neural encoder learns the optimal mapping between attempted and imagined speech neural embeddings. As a baseline, we apply PCA to neural activity, averaging embeddings across time and trials to obtain word-level embeddings, and visualize the top two PCs. For *BIT-All*, we apply PCA to neural encoder outputs and project word-level embeddings onto the top two PCs. These embeddings are taken *after sentence-level fine-tuning* to ensure they contain semantic information. For each word, we use Euclidean distance to quantify similarity between attempted and imagined embeddings (before PCA).

Figure 4B shows a clear separation between attempted and imagined speech projections of neural activity. Since PCA focuses only on neural variance, this separation reflects differences unrelated to sentence content. In contrast, BIT embeddings show much less separation (Figure 4C), indicating that BIT aligns neural embeddings across tasks after accounting for semantic information. Color intensity in Figure 4B-C indicates that embeddings from both tasks show higher similarity after pretraining. In addition, we fit a linear discriminant analysis (LDA) to find the projection that maximizes between-task separation while minimizing within-task variance. We then plot this discriminant axis as a low-dimensional summary of task separability (Appendix L.2). This shows that task embeddings are linearly separable in original neural activity but not after alignment with BIT (Figure 4B-C).

To understand how neural embeddings relate to individual words over time, we use projectors that enable visualization of token-level interactions. While we use an MLP projector in our experiments,

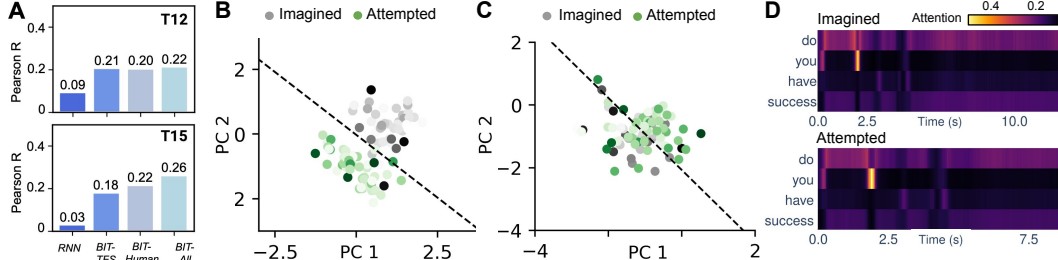

Figure 4: *BIT aligns attempted and imagined speech neural embeddings to enable cross-task generalization.* **(A)** Representational similarity analysis (RSA) scores between neural and audio-LLM text embeddings. **(B)** PCA embeddings of neural features from participant T12 are visualized on the first two PCs. Word-level embeddings are averaged across time and trials and shown as dots. The same words are shown for both tasks. Colors indicate tasks, and color intensity represents the distance between attempted and imagined speech embeddings for each word (darker colors indicate higher similarity). The line shows the LDA linear discriminant. **(C)** For *BIT-All*, PCA is applied to neural encoder outputs from participant T12, with word-level embeddings from the top two PCs visualized as dots. Same plotting conventions as panel B. **(D)** Using a cross-attention projector in BIT allows us to visualize attention weights, which reveal that neural-text temporal alignment is similar across tasks.

linear and cross-attention projectors yield similar, slightly lower performance (Appendix K). We visualize attention weights of a cross-attention projector, showing that embeddings from both tasks exhibit similar neural-text temporal alignment (Figure 4D). These results further confirm that BIT can preserve semantic information to enable cross-task generalization.

## 6 DISCUSSION

**Summary:** In this work, we introduce an end-to-end speech decoding framework. Our method combines a transformer neural encoder, pretrained on human and monkey Utah array recordings across speech and arm-related motor tasks, with either an n-gram LM in a cascaded framework or an audio-LLM for end-to-end optimization. In the cascaded setting, we show that cross-species, cross-task pretraining enables transfer to attempted and imagined speech after fine-tuning, achieving first place on the Brain-to-Text '24 and '25 benchmarks. Integrated end-to-end with Aero1-Audio 1.5B, an audio-extended Qwen2.5-1.5B, BIT reduces WER by over 50% compared to the previous best method. Finally, we demonstrate that BIT learns representations that are shared across imagined and attempted speech, enabling cross-task generalization.

**Limitations and future directions:** Our approach faces several challenges and opportunities for improvement. First, the end-to-end approach requires about 0.95 seconds per sentence on average, which is slower than the cascaded approach (0.24 seconds) and inefficient for real-time BCIs. In addition, the neural encoder is trained with bidirectional attention to maximize performance, rendering it unsuitable for online decoding; switching to causal attention is feasible, albeit at some cost to decoding accuracy. Similarly, while we employ a compact 1.5B audio-LLM, larger models cannot run on-device, further limiting real-time applications. Another consideration is pretraining: we find that human data yield larger gains than cross-species transfer, likely because human datasets include tasks more relevant to speech (and, to a lesser extent, handwriting), whereas monkey reaching data are less applicable. Additionally, the neural encoder requires substantial unlabeled datasets to address sensor variability, and LLMs need large amounts of labeled data to outperform cascaded approaches. However, we have limited access to private human data. Finally, the LLM decoder itself can be improved through better modality alignment and prompt design, and there remains a broader need to develop methods addressing non-stationarity (Le et al., 2025), neural plasticity, and user-interface co-adaptation for long-term BCI use.

**Broader impact:** This work improves current cascaded speech BCI systems to restore high-accuracy communication for users with paralysis. It also advances SOTA end-to-end models, making them easier to optimize and deploy in real-time applications. These developments could enable patients who are unable to communicate or move to interact with the outside world and perform everyday tasks with AI assistance. At the same time, it raises important safety and privacy concerns, as decoding inner speech must require explicit user consent and reliable safeguards (Kunz et al., 2025).

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

## A    DATASET DETAILS

This section documents the studies and sources of all datasets used for pretraining, as well as for attempted and imagined speech decoding, and provides a brief description of each.

### A.1    HUMAN DATA

**Willett et al. (2021).** This dataset contains recordings from a participant with hand paralysis who attempted and imagined handwriting, decoded in real time by an intracortical BCI. Data are publicly available at the DRYAD repository. Only threshold crossings from 192 electrodes are available.

**Willett et al. (2023).** This dataset is associated with the Brain-to-Text Benchmark '24 and comes from a study in which researchers developed a speech BCI that decodes neural activity from intracortical microelectrode arrays to reconstruct speech in real time for a participant with amyotrophic lateral sclerosis (ALS). The dataset contains sentences with both a 50-word vocabulary and a 125,000-word vocabulary. Neural recordings during attempted speech are publicly available at the DRYAD repository. Both threshold spikes and spike-band power (SBP) are provided from 256 electrodes. However, we use only 128 of them for decoding, as we find that this empirically leads to the best performance.

**Fan et al. (2023).** This dataset comes from a study in which researchers introduced a framework allowing a speech BCI to self-recalibrate without interrupting the user. Neural recordings are publicly available at the DRYAD repository, with only threshold crossings available from 192 electrodes.

**Karpowicz et al. (2024).** This dataset is from the Few-shot Algorithms for Consistent Neural Decoding (FALCON) Benchmark, which evaluates intracortical BCI decoders for stability across sessions. We use only the human handwriting subset, also associated with Fan et al. (2023), available at DANDI. The threshold spike data in this subset also include 192 electrodes.

**Card et al. (2024).** This dataset is associated with the Brain-to-Text Benchmark '25 and comes from a study in which a speech BCI decoded neural activity from the left precentral gyrus of a person with ALS and severe dysarthria. Neural recordings during attempted speech, containing a 125,000-word vocabulary, are publicly available at the DRYAD repository, with both threshold spikes and SBP available from 256 electrodes.

**Kunz et al. (2025).** This dataset comes from a study in which a speech BCI decodes imagined speech from motor cortex activity to enable real-time communication without vocalization. It includes recordings and sentences from four participants during attempted, imagined and perceived speech tasks, publicly available at the DRYAD repository, with both threshold spikes and SBP from 256 electrodes. For pretraining, we use all four participants, but decoding uses only T12 and T15.

### A.2    NON-HUMAN PRIMATES (NHP) DATA

**Churchland et al. (2012).** This dataset comes from a study in which neural population activity in primary motor cortex (M1) was recorded from non-human primates (NHPs) during center-out reaching tasks. It includes thresholded spikes from 192 electrodes, publicly available at DANDI.

**O'Doherty et al. (2017).** This dataset originates from a study in which researchers recorded neural activity from the sensorimotor cortex of NHPs during self-paced reaching tasks. It includes thresholded spikes from two monkeys, recorded from 192 electrodes. The data are publicly available at the Zenodo repository.

**Perich et al. (2018).** This dataset is from a study in which researchers recorded neural activity from M1 and dorsal premotor cortex (PMd) of four macaque monkeys during reaching tasks. Neural data are publicly available at the DANDI repository with thresholded spikes. The data contains spike sorted activity in three subjects, and threshold crossings in the fourth subject from 192 electrodes.

**Chowdhury et al. (2020).** This dataset comes from a study in which researchers recorded neural activity from area 2 of the primary somatosensory cortex in rhesus macaques during reaching tasks. It includes thresholded spike data from 92 channels, publicly available at the DRYAD repository.

**Even-Chen et al. (2019).** This dataset originates from a study in which researchers recorded neural activity from PMd in rhesus macaques during an instructed-delay center-out reaching task. It includes thresholded spike data from 96-channel Utah arrays, publicly available at the DANDI repository.

**Ma et al. (2023).** This dataset is from experiments in which researchers recorded neural activity from M1 in rhesus macaques across many days while the animals performed various motor tasks (wrist force, grasping, manipulandum reaching, etc.). They used thresholded multi-unit spike data from 96-channel microelectrode arrays. The dataset is publicly available at the DRYAD repository.

**Temmar et al. (2025).** This dataset comes from a study in which researchers recorded neural activity from M1 in a rhesus macaque performing a self-paced finger movement task. It includes thresholded multiunit spikes from 96-channel Utah microelectrode arrays, together with behavioral measurements of finger positions and velocities. The dataset is available at the DANDI repository.

### A.3 DATA PREPROCESSING

For all datasets, we resample neural activity into 20 ms time bins (if not already provided at this resolution) and apply z-scoring across days to mitigate non-stationarity. Without this normalization, speech decoding WERs increase substantially due to severe data drift. In addition, when SBP is available, we include it in decoding, as we find empirically that SBP contributes more to decoding accuracy than thresholded spikes alone. Combining thresholded spikes and SBP leads to the lowest WER (Table 3). Human datasets generally contain higher-quality recordings, while some monkey datasets include dead channels. If fewer than two dead channels are detected, we interpolate them with the mean neural activity over time, though we note that more careful handling would be preferable. Datasets with more than two dead channels are excluded from pretraining.

| Participant | Thresholded | SBP | Thresholded + SBP |
|---|---|---|---|
| T12 | 21.35% | 18.64% | 17.26% |

Table 3: *Ablation of features used for speech decoding.* Validation PER is reported.

Regarding data quality, we also considered other human and monkey datasets not listed but ultimately excluded them from pretraining, as their inclusion led to reduced pretraining performance. This may be due to data quality issues that require more careful examination in future work, which is beyond the scope of the present study.

### A.4 DATA SPLITS FOR PRETRAINING

As SSL pretraining does not rely on labels, all neural data from the datasets can be incorporated into the training set, including the evaluation (validation) and holdout datasets provided by the Brain-to-Text '24 and '25 benchmarks. However, since our primary interest is evaluating whether the transformer effectively captures the neural representations of participants T12 and T15, we use the validation datasets from these participants to select the optimal model checkpoint, based on the metric for neural activity reconstruction quality.

## B METRICS

$R^2$ **for neural activity reconstruction.** We use the coefficient of determination ($R^2$) as the metric to evaluate neural activity reconstruction quality during SSL pretraining. The $R^2$ score measures the proportion of variance in the target neural activity that can be explained by the model's predictions, with higher values indicating better reconstruction. During pretraining, the validation $R^2$ is used to select the model checkpoint for downstream speech decoding.

**PER.** We use a 41-token phoneme vocabulary, including all phonemes plus silence and the CTC blank token. The phoneme error count is first computed by removing repeats and blanks from the decoded phoneme sequences, then calculating the Levenshtein edit distance (Levenshtein, 1966) to the ground-truth sequences, counting substitutions, insertions, and deletions. The phoneme error rate is then obtained by normalizing the error count by the total number of ground-truth phonemes.

**WER.** Word error rate is computed similarly to PER. Blank and special tokens are removed from the decoded sentences to obtain word sequences. The Levenshtein edit distance between the predicted and ground-truth word sequences is then calculated, counting substitutions, insertions, and deletions. WER is obtained by normalizing the total error count by the number of words in the ground-truth sequences.

## C  PHONEME-LEVEL DECODING

For brevity, intermediate phoneme decoding results are omitted from the main text. Nonetheless, we observe that neural encoders achieving lower PERs consistently lead to lower WERs, whether combined with a 5-gram LM in the cascaded framework or with a LLM decoder in the end-to-end framework. In the main text, we present the holdout or validation WERs for each baseline neural encoder; in this section, we provide the validation PERs for the baseline encoders, as summarized in Table 4.

| Neural Encoder | Attempted Speech | | Imagined Speech | |
|---|---|---|---|---|
| | T12 | T15 | T12 | T15 |
| RNN | 18.67% | 9.64% | 30.81% | 24.56% |
| BIT-TFS | 17.26% | 8.87% | 25.01% | 21.21% |
| BIT-Human | 15.95% | 7.61% | 19.63% | 18.83% |
| BIT-All | 14.39% | 7.12% | 18.08% | 17.94% |
| BIT-Cross-Task-Only | – | – | 20.46% | 19.58% |

Table 4: *Phoneme decoding benchmark.* The metrics shown are the validation PER.

In the end-to-end model, phonemes serve as intermediate targets that encode phonetic information into neural representations to guide LLM fine-tuning, even when the final goal is sentence prediction. We observe that, without phoneme-decoding training, simply pretraining the neural encoder on a large data corpus does not lead to low WERs during the sentence-level decoding stage, highlighting the importance of phonetic information for speech decoding tasks. This phenomenon is also reported in previous end-to-end methods (Feng et al., 2024), which employ a two-stage decoding strategy that trains the model for phoneme prediction before sentence prediction.

## D  CASCADED DECODER

The cascaded decoder consists of a 5-gram LM and an OPT model for re-scoring the LM outputs. In this section, we briefly describe the resources and procedures used; for comprehensive details, readers are referred to the original studies (Willett et al., 2023; Card et al., 2024).

**5-gram LM.** We use the 5-gram LM provided by the Brain-to-Text Benchmark '24 and '25 (Willett et al., 2023; Card et al., 2024) to decode word sequences from neural encoder outputs. The pretrained model is publicly available on DRYAD. The 5-gram LM is created using OpenWebText2 corpus and therefore has a large vocabulary, constructed from 634 million sentences with 99 billion words. The LM decoder performs an approximate Viterbi (beam) search, which maps CTC phoneme label sequences from the encoder to words. During inference, the beam search integrates state transition probabilities with the encoder's predicted CTC phoneme label probabilities to find the most likely word sequences. See Section 8 of the supplementary materials for details on constructing the 5-gram LM and performing inference with it. In Table 5, we report the parameters used for beam search with a 5-gram LM.

**OPT re-scoring.** Given the sentence hypotheses from the 5-gram LM, an OPT model (Zhang et al., 2022) is employed to rescore the candidate sentences and further improve decoding accuracy. A publicly available pretrained OPT with 6.7B parameters is used to rescore the n-best outputs from the n-gram LM. The re-scoring procedure considers both the probability of the sentence's corresponding CTC phoneme label sequence output by the encoder (*acoustic score*) and the sentence probability estimated by the n-gram LM; see Section 8.5 of the supplementary materials for details on the equation used for OPT re-scoring and the LM decoding parameters. In Table 5, we report the parameters used for language model re-scoring with OPT.

| Hyperparameter | Value |
|---|---|
| Beam Size (Number of Candidate Sentences) | 100 |
| Acoustic Score Scale Factor | 0.325 |
| Blank Penalty | 90 |
| OPT Re-Score Scale Factor ($\alpha$) | 0.55 |

Table 5: Hyperparameters used for inference with the cascaded decoder (5-gram LM with beam search + OPT sentence re-scoring).

## E END-TO-END DECODER

**Text-based LLMs.** We include models from the Qwen2.5 and Qwen3 families. ***Qwen2.5-1.5B*** (Qwen et al., 2025) is a mid-sized LLM trained with an improved pretraining pipeline and instruction tuning. We also include ***Qwen2.5-7B***. To examine the effect of model size and architectural advances, we include two recent models from the Qwen3 series: ***Qwen3-0.6B*** (Yang et al., 2025), a compact model optimized for efficiency, and ***Qwen3-1.7B***, a larger variant designed to provide stronger language modeling capacity while remaining computationally tractable. Together, these models provide a representative set of text-based LLMs at approximately 1B scale, enabling comparison with audio-augmented counterparts under similar parameter regimes.

**Audio-based LLMs.** In addition to text-only baselines, we consider audio-augmented LLMs. ***Aero1-Audio 1.5B*** (Li et al., 2025) extends Qwen2.5-1.5B with an audio front-end, enabling the model to process acoustic representations alongside text. To understand the effect of model scale in the audio domain, we also include ***Qwen2-Audio 7B*** (Chu et al., 2024), a larger audio-based LLM capturing richer acoustic and semantic features. Comparing Aero1-Audio 1.5B and Qwen2-Audio 7B with text-only models of similar size allows us to quantify the contributions of audio conditioning relative to purely textual modeling in speech decoding from neural activity.

## F LLM INFERENCE FOR END-TO-END DECODING

Text generation uses nucleus (top-$p$) sampling with a parameter of $p = 0.9$ and a temperature of 0.7, generating up to 25 new tokens per sequence. Top-$p$ sampling restricts token selection to the smallest set of tokens whose cumulative probability exceeds $p$, focusing on high-likelihood tokens while allowing some diversity. The temperature controls randomness: lower values produce more confident, deterministic outputs, while higher values increase variability. Although more advanced strategies such as beam search can be used, they significantly increase inference time when the number of beams exceeds five. Therefore, we use nucleus sampling for efficiency. We acknowledge that better LLM inference strategies could further improve end-to-end decoding performance, but this is beyond the scope of the present work.

## G MODEL ENSEMLBING USING LLM MERGER

To further improve the decoding accuracy of the cascaded and end-to-end decoders for benchmark submissions, we employ a model ensembling approach with LLM merging (Li et al., 2024). While the 5-gram and OPT models can partially correct phoneme decoding errors, their outputs are not always perfect, and different neural encoders produce varying sentence candidates. To mitigate these errors, a fine-tuned LLM can be used to evaluate multiple sentence candidates from an ensemble of cascaded or end-to-end encoders. Using both the transcription candidates and their corresponding phoneme sequences as inputs, Li et al. (2024) finetuned an LLM to generate the correct sentence. This setup enables the model to learn the relationship between predicted phonemes and decoded sentences, as well as to identify common, model-specific errors across decoder outputs. For both cascaded and end-to-end approaches, we provide the candidate sentences along with the phoneme outputs from neural encoders that are randomly initialized with different seeds to the fine-tuned LLM, which then generates the final best sentence used for evaluation.

## H BRAIN-TO-TEXT BENCHMARK '24 AND '25 DETAILS

**Brain-to-Text Benchmark '24.** The top scores on the leaderboards for both cascaded and end-to-end methods are achieved using model ensembling with LLM merging. For this benchmark, we initialize 10 cross-species, cross-task pretrained encoders with different random seeds, each fine-

tuned to decode sentences from participant T12 data. Each model is trained on the training data provided by the benchmark and uses the provided validation set for checkpoint selection. Each of the 10 models outputs its own phoneme and sentence sequences, which are then provided to a fine-tuned GPT-3.5 to select the final best sentence sequence. This procedure is applied to both the entries *BIT Cascaded + Ensemble* and *BIT End-to-End + Ensemble*, which are the top-ranked submissions for the cascaded and end-to-end approaches on the leaderboard. All of our other leaderboard entries are based on a single model with a single seed. For information on the other leaderboard entries, see Willett et al. (2024).

**Brain-to-Text Benchmark '25.** The top scores on the public leaderboards for both cascaded and end-to-end methods are achieved using model ensembling with LLM merging. For this benchmark, we initialize 29 cross-species, cross-task pretrained encoders with different random seeds, each fine-tuned to decode sentences from participant T15 data. We fine-tune GPT-4 to merge the phoneme and sentence outputs from these 29 models. All of our other leaderboard entries are based on a single model with a single seed. In Table 2, the *RNN (baseline)* uses a single RNN encoder with a 5-gram LM, labeled "UCD-NPL causal RNN + 5gram." The *RNN + Ensemble* combines multiple RNNs with LLM merging, labeled "Stanford-NPTL causal RNN Ensemble + 5gram." The *RNN-TTA + Pseudo-Ensemble* generates diverse outputs from a single RNN model via noise injection and selects the final sentence with an LLM merger; it is listed as "Stanford-NPTL causal RNN TTA-Ensemble + 5gram," where TTA stands for test-time adaptation.

## I    CONTRASTIVE LEARNING FOR MODALITY ALIGNMENT

We align neural and text embeddings in a shared latent space using contrastive learning. Let $\mathbf{z}_i^s \in \mathbb{R}^{T \times D}$ and $\mathbf{z}_i^t \in \mathbb{R}^{L \times D}$ denote the neural and text embeddings, respectively, for the $i$-th sample in a batch of $B$ trials. Here, the neural embeddings are obtained by projecting the neural encoder outputs through the MLP projector into the text embedding space. Each of $\mathbf{z}_i^s \in \mathbb{R}^{T \times D}$ and $\mathbf{z}_i^t \in \mathbb{R}^{L \times D}$ is first mean-pooled to a single vector by averaging across its sequence dimension ($T$ and $L$), then projected into a shared latent space of dimension $P$ via modality-specific linear layers, followed by $\ell_2$ normalization. We denote the resulting modality-specific vectors as $\tilde{\mathbf{z}}_i^s \in \mathbb{R}^P$ and $\tilde{\mathbf{z}}_i^t \in \mathbb{R}^P$, which we refer to as the "modality tokens."

Contrastive learning is then applied to pull together corresponding neural and text modality tokens (positive pairs) while pushing apart tokens from different samples (negative pairs) within a training batch. Following a symmetric InfoNCE loss (Oord et al., 2018; Radford et al., 2021), the loss for a batch is defined as

$$\mathcal{L}_{\text{contrastive}} = \frac{1}{2B} \sum_{i=1}^{B} \left[ -\log \frac{\exp(\tilde{\mathbf{z}}_i^s \cdot \tilde{\mathbf{z}}_i^t / \tau)}{\sum_{j=1}^{B} \exp(\tilde{\mathbf{z}}_i^s \cdot \tilde{\mathbf{z}}_j^t / \tau)} - \log \frac{\exp(\tilde{\mathbf{z}}_i^t \cdot \tilde{\mathbf{z}}_i^s / \tau)}{\sum_{j=1}^{B} \exp(\tilde{\mathbf{z}}_i^t \cdot \tilde{\mathbf{z}}_j^s / \tau)} \right],$$

where $\tau$ is a learnable temperature parameter that scales the cosine similarities.

In our experiments, we apply a learnable scaling factor to the similarity logits when computing the contrastive learning objective. This factor is parameterized by a learnable temperature, initialized at 0.1 and constrained to a maximum value of 100. Due to the large size of the LLMs and limited computational resources, we use a small batch size (Table 13) for contrastive learning. While this batch size enables measurable performance gains in the current study, it may limit the effectiveness of contrastive learning, which typically benefits from larger batch sizes to better sample negative pairs. Future work should explore larger batch sizes to fully realize the potential of contrastive learning and further improve cross-modal alignment.

## J    LOSS FUNCTION FOR BIT

We train the end-to-end model with cross-entropy loss for neural-to-text translation and contrastive loss for neural–text alignment. Let $\mathbf{y}_i = (y_{i,1}, \ldots, y_{i,L_i})$ denote the ground-truth token sequence for the $i$-th sample in a batch of size $B$. The predicted probability distribution over a vocabulary of size $V$ at the $t$-th token is denoted as $\hat{\mathbf{p}}_{i,t} = f\big(\text{concat}(\mathbf{z}_i^s, \mathbf{z}_i^t)\big) \in \mathbb{R}^V$, where $f$ represents the LLM decoder, and the neural and text tokens are concatenated along the sequence dimension. The

cross-entropy loss for the batch is then defined as

$$\mathcal{L}_{\text{CE}} = -\frac{1}{B} \sum_{i=1}^{B} \sum_{t=1}^{L_i} \log \hat{p}_{i,t}(y_{i,t}).$$

The total training objective is given by $\mathcal{L}_{\text{BIT}} = \mathcal{L}_{\text{CE}} + \mathcal{L}_{\text{contrastive}}$.

## K   PROJECTOR FOR LLM DECODER

To project neural embeddings into the LLM text embedding space, we explore different projectors, including linear, MLP, and cross-attention modules. We benchmark these projectors to evaluate their impact on end-to-end decoding performance. Results are summarized in Table 6.

| Projector Variant | Imagined Speech (T12) | Imagined Speech (T15) |
|---|---|---|
| BIT (Linear Projector) | 19.47% | 18.09 % |
| BIT (Cross-Attention Projector) | 17.33% | 17.67% |
| BIT (MLP Projector) | 16.39% | 13.61% |

Table 6: *Ablation of the projector in BIT.* The metrics shown are validation WER.

**Cross-attention projector.** This module applies cross-attention from neural tokens (queries) to text tokens (keys and values), enabling token-level interactions between modalities. Input neural and text embeddings, $\mathbf{z}^s$ and $\mathbf{z}^t$, are first normalized with layer normalization and projected linearly to a shared hidden dimension $d_{\text{hidden}}$. Multi-headed attention is then applied, followed by a learnable residual connection to stabilize training. The attended outputs are finally projected back to the text embedding dimension $d_{\text{text}}$. Formally, the module computes

$$\mathbf{z}^*, \mathbf{A} = \text{CrossAttentionProjector}(\mathbf{z}^s, \mathbf{z}^t),$$

where $\mathbf{z}^*$ are the aligned neural-text representations and $\mathbf{A}$ are the attention weights (Figure 4D). The architecture includes tunable parameters such as the hidden dimension $d_{\text{hidden}}$, number of attention heads $n_{\text{heads}}$, drop out ratio, and a learnable residual scaling factor.

| Hyperparameter | Value |
|---|---|
| Hidden Dimension ($d_{\text{hidden}}$) | 256 |
| Number of Heads ($n_{\text{heads}}$) | 1 |
| Learnable Residual Scaling Factor | 0.5 |
| Dropout Ratio | 0.1 |

Table 7: Hyperparameter used for the cross-attention projector.

## L   INTERPRETABILITY ANALYSIS

### L.1   REPRESENTATIONAL SIMILARITY ANALYSIS

We restrict the analysis to sentences with neural time-patch token lengths between $45 \sim 80$ for participant T12 and $120 \sim 200$ for participant T15, ensuring sufficient temporal resolution and balanced coverage (see Figure 5). Neural embeddings are converted into fixed-length vectors using segmented mean pooling, where each sequence is divided into 10 segments and the segment averages are concatenated to preserve coarse temporal dynamics. In contrast, LLM embeddings are reduced to sentence vectors via mean pooling across tokens. For each modality (neural and text), we compute a representational dissimilarity matrix (RDM) (Kriegeskorte et al., 2008) using one minus the cosine similarity. To compare representational structures, we extract the upper-triangular entries of each RDM and compute the Pearson correlation coefficient between neural and LLM RDMs. The resulting RSA score quantifies how well the geometry of neural embeddings aligns with language structures in LLMs. Framing RSA as an interpretability tool, this analysis demonstrates that the transformer encoder learns representations more consistent with language-level structure than RNNs. RSA is applied only to attempted speech data from participants T12 and T15 in the studies (Willett et al., 2023; Card et al., 2024).

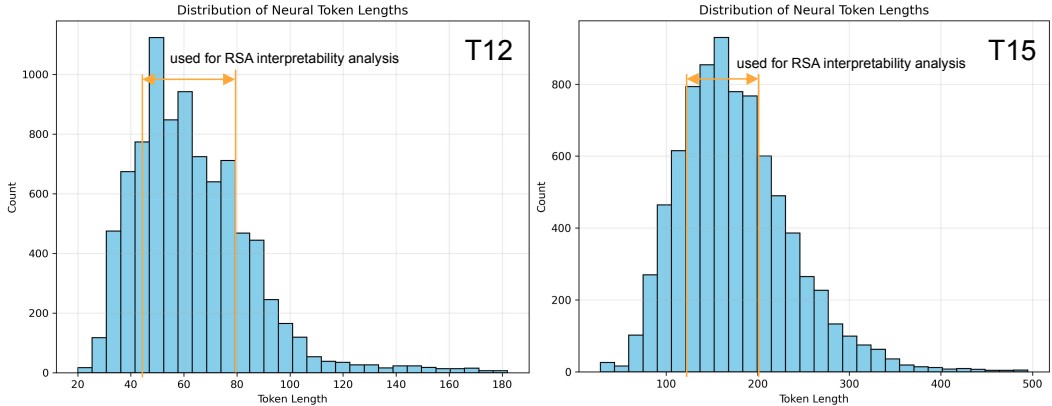

Figure 5: *Distribution of neural token lengths across sentences for RSA.* We restrict RSA to sentences with token lengths between 45 and 80 (mean length ≈ 63) for participant T12 and between 120 and 200 (mean length ≈ 160) for participant T15, since neural embeddings are converted into fixed-length sentence vectors by dividing each sequence into ten temporal segments and concatenating their averages. Sequences that are too short lack sufficient resolution, while overly long sequences introduce imbalance. These ranges ensure reliable and comparable sentence-level embeddings for cross-modal RSA.

## L.2 LDA and Embedding Distances

We fit an LDA using *scikit-learn*. For each word, neural embeddings are first projected onto the PCA subspace and then averaged across time and trials to produce a single feature vector. The resulting low-dimensional neural projections are used as features to fit the LDA to predict trial type (attempted vs. imagined speech for binary classification). We restrict the LDA to a single dimension to identify an axis that linearly separates neural projections from the two tasks for visualization purposes. However, when computing Euclidean distances between attempted and imagined speech embeddings, we use the original high-dimensional embeddings for each word before PCA projection to ensure comparisons reflect both direction and magnitude in the embedding space. These analyses are applied only to attempted and imagined speech data from participants T12 and T15 in the study (Kunz et al., 2025).

## M    Effects of supervised cross-subject pretraining on decoding

We investigated whether supervised cross-subject pretraining could enhance downstream speech decoding performance. Incorporating data from multiple subjects during phoneme- and sentence-level decoding led to significantly worse performance compared to single-subject training. This performance degradation may arise from inter-subject functional variability (differences in how neural activity relates to behavior) rather than sensor variability (e.g., electrode placement or noise), as SSL cross-subject pretraining does improve performance after fine-tuning on single-subject data. These results indicate that, with the current model architectures and datasets, supervised cross-subject pretraining does not improve performance, and more sophisticated modeling strategies for addressing inter-subject functional variability will be needed in future work.

## N    Decoding Error Analysis

### N.1    Phoneme-Level Decoding Errors of the Neural Encoder

We performed a detailed decoding error analysis to characterize model behavior at the phoneme level. For each sentence, we first tokenized the predicted and ground truth phoneme sequences, then computed the edit distance. To study systematic substitution patterns, we aligned every predicted sequence with its corresponding ground truth using a dynamic programming algorithm that explicitly models match, substitution, deletion, and insertion operations. Insertions and deletions were represented with a dedicated null token so that both sequences remain length matched after alignment.

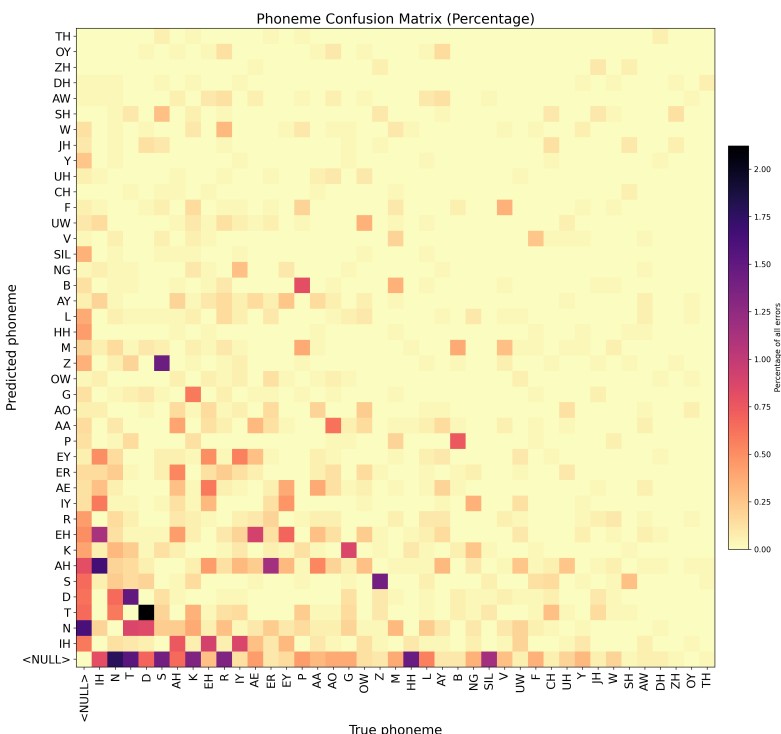

Figure 6: *Phoneme-level decoding error matrix.* Predicted and ground truth phoneme sequences are aligned, correct matches are removed, and all remaining errors are normalized to percentages. Phonemes are ordered by total error frequency, with true phonemes on the horizontal axis and predicted phonemes on the vertical axis. The visualization highlights dominant substitution patterns and systematic decoding errors.

Based on the aligned pairs, we aggregated all observed phoneme confusions across the validation set. For every occurrence of a true phoneme aligned with a predicted phoneme, we incremented a global confusion counter. This produced a full phoneme by phoneme confusion matrix that includes substitutions, deletions, and insertions. To emphasize the structure of decoding errors, we sorted phonemes by their total off diagonal error counts. The matrix was then visualized with the most error prone phonemes placed at the bottom of the vertical axis and the right side of the horizontal axis, which makes high error regions visually salient.

Inspection of the matrix further reveals several prominent confusion clusters. As show in Figure 6, /D/ and /T/ show frequent reversals, /B/ and /P/ exhibit substitution, and /S/ often confuses with /Z/. In addition, the central vowels /AH/, /IH/, and /EH/ are commonly misdecoded due to their acoustic proximity. Overall, the observed error structure matches known phonetic similarity patterns and appears consistent with expected decoder behavior.

## N.2 WORD-LEVEL DECODING ERRORS OF THE LLM DECODER

To examine word level decoding errors, we first computed the full confusion matrix across all predicted and true words. Figure 7-a shows the complete matrix. Because it includes every word arranged alphabetically, the matrix is very large and individual error pixels can only be seen at extremely high resolution. Even so, the matrix exhibits a clear diagonal structure, indicating that the model achieves high word level decoding accuracy. Most predictions fall near the correct location, and the observed errors tend to occur between words that are similar in spelling or phonological form.

At the same time, we note that many errors occur only once. These sparse, non recurrent mistakes behave more like noise or one off events rather than consistent confusion patterns. To highlight the relatively more frequent error types, we filtered the matrix to retain only word pairs that appeared in at least two errors. After this filtering, the total number of error events dropped from 734 to 58. The

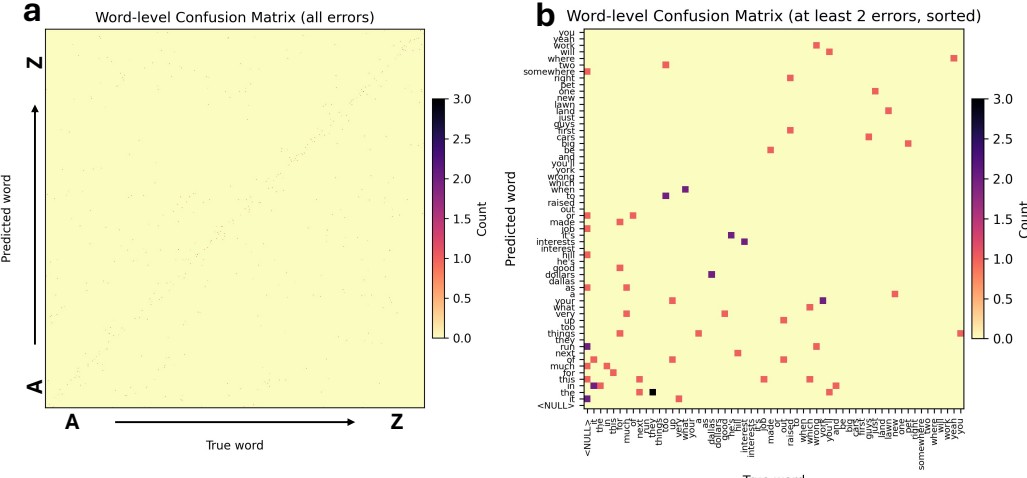

Figure 7: *Word-level decoding error matrix.* **(a)** Word level confusion matrix computed over all decoding errors. Words are arranged alphabetically from a to z for both axes. Because the vocabulary is large, the full matrix must be viewed at a very large scale to see individual pixels clearly. A clear diagonal structure appears, indicating that most decoding mistakes occur between words with similar spellings or phonological forms. **(b)** Word level confusion matrix after filtering to retain only word pairs that exhibit at least two errors. This highlights the more systematic confusions and removes the many near zero entries present in panel a. The resulting matrix reveals the dominant recurrent error patterns more clearly.

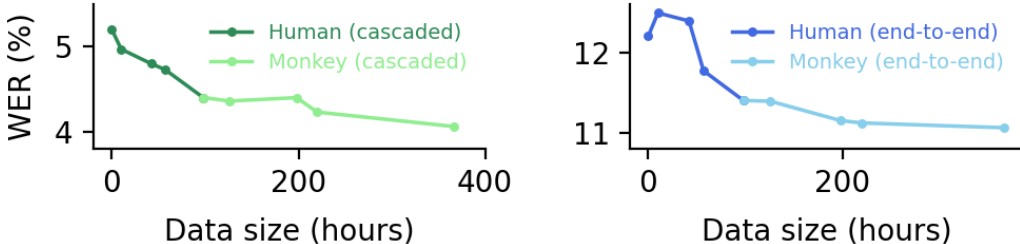

Figure 8: *Impact of progressively increasing the proportion of human versus monkey pretraining data on attempted-speech decoding performance for participant T15.* Across both cascaded and end-to-end models, "human" corresponds to the fraction of human pretraining data progressively added, whereas "monkey" indicates the fraction of monkey pretraining data added on top of the human data.

filtered matrix is shown in Figure 7-b. Although the maximum error count after filtering is still only three, which indicates that the model does not develop highly stable confusion patterns at the word level, these repeated errors at least reveal small but observable tendencies in a handful of word pairs.

Overall, these analyses show that the model performs well at word level decoding. Errors are limited in number and largely reflect marginal or near neighbor confusions. Even the repeated errors occur only in a very small subset of word pairs, underscoring the generally high decoding accuracy.

## O   SCALING CURVE

We examine how gradually increasing the amount of human versus monkey pretraining data affects attempted speech decoding performance for participant T15. In the human-only condition, we expand the pretraining corpus by progressively adding human Utah-array datasets, starting from no human data and reaching the full 97.8 hours. In the cross-species condition, we begin with the complete human corpus and then incrementally incorporate monkey Utah-array datasets, ultimately extending the total pretraining duration to 366.8 hours across both species. Figure 8 shows how decoding performance changes as these proportions vary, and highlights that adding human data yields larger performance gains than augmenting the model with monkey data.

## P  ABLATION OF ENCODER PRETRAINING

Our pretraining procedure includes the human neural data used later for finetuning, but discards all labels. Whether to include such datasets during pretraining depends on the type of generalization being evaluated: excluding them tests generalization to unseen data, whereas including them tests generalization to unseen tasks. Because our objective is to improve transfer to speech-decoding tasks, we choose to include the available human speech datasets during pretraining. This also allowed us to make full use of the limited human data to help stabilize the learned representations. To test generalization to entirely unseen data, however, we also repeated pretraining while excluding all human speech datasets. As shown in Table 8, including or excluding the human neural data used for finetuning leads to no substantial difference in attempted speech decoding performance.

|  | (a) Cascaded model | | (b) End-to-end model | |
| --- | --- | --- | --- | --- |
|  | Finetune data included | Finetune data excluded | Finetune data included | Finetune data excluded |
| T12 | 6.35% | 7.24% | 15.67% | 15.34% |
| T15 | 4.06% | 4.36% | 11.06% | 11.00% |

Table 8: *Impact of including versus excluding fine-tuning datasets during pretraining on attempted speech decoding performance.* Excluding the fine-tuning datasets tests generalization to unseen data, whereas including them tests generalization to unseen tasks.

In Figure 2, the comparison between *BIT-All* and *BIT-Cross-Task-Only* for imagined speech decoding does not control for differences in data size, as our goal was to include additional data beyond same-participant data to fully benefit from SSL pretraining. To control for data size differences, we also conducted an experiment comparing SSL and supervised pretraining using equal data sizes, with results reported in Table 9. When controlling for data size, we find no substantial performance difference between SL and SSL pretraining for imagined speech decoding.

|  | T12 (Cascaded) | T12 (End-to-End) |
| --- | --- | --- |
| BIT-Cross-Task-Only | 12.53% | 15.71% |
| BIT-SameParticipant-SSL | 12.67% | 15.64% |

Table 9: *Impact of SL versus SSL pretraining using equal amounts of human speech data on imagined speech decoding performance. BIT-Cross-Task-Only is pretrained using SL, whereas BIT-SameParticipant-SSL is pretrained using SSL, and both use the same human speech data from participant T12.*

## Q  MODEL AND HYPERPARAMETER DETAILS

**Transformer Encoder.** We implement a transformer encoder with an architecture similar to Feghhi et al. (2025). Since their official code is not publicly available at the time of this submission, we construct our own architecture using custom model parameters listed in Table 10. To select the optimal model architecture and optimizer configuration for training on single-subject data (T12 and T15), we use *Ray Tune* (Liaw et al., 2018) to randomly sample 30 optimizer hyperparameter (batch size, weight decay, and learning rate) combinations from the ranges listed in Table 12. The optimizer hyperparameters for single-subject phoneme-level decoding is selected based on these ranges. For imagined speech decoding, we reuse the same model hyperparameters for both participants. For pretraining, we reuse the same model architecture while adjusting the learning rate, weight decay and batch size to $5 \times 10^{-4}$, $1 \times 10^{-5}$ and 64, respectively. We do not increase the model depth when scaling to human and monkey datasets, as SSL pretraining performance (evaluated using the validation $R^2$ metric for neural activity reconstruction) shows no decline, indicating that the model capacity is sufficient to handle the data variability. The transformer model comprises $\sim 7$ million parameters. The inclusion of subject-specific patch embedding modules and linear decoders increases the total number of model parameters to $\sim 13$ million.

**RNN Encoder.** For the RNN encoder, we use the PyTorch implementation provided by the benchmark host in the official GitHub repository. We retain the original model architecture (Table 11) and apply *Ray Tune* (Liaw et al., 2018) to randomly sample 30 optimizer hyperparameter (batch size, weight decay, and learning rate) combinations from the ranges listed in Table 12, using attempted

| Hyperparameter | Value |
|---|---|
| Embedding Dimension | 384 |
| Head Dimension | 512 |
| Number of Heads | 6 |
| Depth | 7 |
| Mask Ratio | 0.5 (T12) and 0 (T15) |
| Max Mask Time Span | 15 |
| Patch Size | 5 |
| Dropout Rate | 0.2 |
| Bidrectional | True |
| Attention Dropout Rate | 0.4 |
| White Noise Standard Deviation | 0.2 |
| Constant Offset Standard Deviation | 0.05 |
| Gaussian smoothing width | 2.0 |

Table 10: Hyperparameters used for the transformer-based encoder.

speech data from participants T12 and T15. For imagined speech decoding, we reuse the same model hyperparameters for both participants, except for the number of day-specific read-in layers, which differs across datasets.

| Hyperparameter | Value |
|---|---|
| Embedding Dimension | 1024 |
| Depth | 5 |
| Mask Ratio | 0.1 |
| Dropout Rate | 0.4 |
| Stride Length | 4 |
| Kernel Length | 32 |
| Number of Day Layers | 24 (T12) and 45 (T15) |
| Bidrectional | True |
| White Noise Standard Deviation | 0.8 |
| Constant Offset Standard Deviation | 0.2 |
| Gaussian smoothing width | 2.0 |

Table 11: Hyperparameters used for the RNN encoder.

| Hyperparameter | Value Range |
|---|---|
| Batch size | [8, 16, 32, 64] |
| Weight Decay | Log-Uniform($5 \times 10^{-5}$, 0.1) |
| Learning Rate | Log-Uniform($5 \times 10^{-5}$, 0.001) |

Table 12: The range of possible optimizer hyperparameters for training the transformer and RNN for phoneme-level decoding, from which *Ray Tune* randomly samples combinations.

**LLM Decoder and LoRA.** We employ the same configuration for the MLP projector and LoRA across all text- and audio-based LLM decoders (see details in Table 13). For text-based LLMs, the MLP hidden dimension is set to match the hidden dimension of the LLM. For audio-LLMs, the MLP hidden dimension is set to match that of the audio encoder's multimodal projector. Depending on the LLM decoder, LoRA is applied to different modules. For all decoders, we apply LoRA to the query, key, value, and output projection layers of the multi-headed self-attention mechanism, as well as the projection layers in the feed-forward MLP. For audio-based Qwen models, we additionally apply LoRA to the linear layer of the multimodal projector. The hyperparameters for cross-modal alignment with contrastive learning are described in the Appendix I.

# R  TRAINING DETAILS

**Transformer Encoder.** For both pretraining and fine-tuning, we train the model using the AdamW optimizer (Loshchilov & Hutter, 2017). Encoders pretrained on human-only or human-and-monkey datasets are trained on a single NVIDIA A100 GPU (80 GB memory) in under 2 days for a total of 400 epochs. Phoneme decoding is performed on a single NVIDIA A40 (48 GB memory) or A100

| Hyperparameter | Value |
|---|---|
| Activation | ReLU |
| Learning Rate | $5 \times 10^{-5}$ |
| Weight Decay | $1 \times 10^{-5}$ |
| Batch Size | 16 (T12) and 8 (T15) |
| Gradient Accumulation Step | 1 (model $<$ 7B) and 8 (model $\geq$ 7B) |
| LoRA Rank | 8 |
| LoRA Scaling Factor | 32 |
| LoRA Dropout Ratio | 0.2 |

Table 13: End-to-end model hyperparameters, including the MLP projector, the LLM decoder and LoRA.

GPU (40 or 80 GB memory) in less than 8 hours for T12 and less than 1 day for T15, for 800 epochs. For hyperparameter tuning, we use 4 NVIDIA A40 GPUs for phoneme decoding, completing the process in less than 2 days. During training, we use validation metrics ($R^2$, PER) to select the best model checkpoint, based on the highest $R^2$ or lowest PER. We employ mixed precision (*bfloat16*) to improve training efficiency for both pretraining and fine-tuning, and all models fit on a single NVIDIA GPU.

**RNN Encoder.** We train the model using the AdamW optimizer. Phoneme decoding is performed on a single NVIDIA A40 (48 GB memory) or A100 GPU (40 or 80 GB memory) in less than 1 day for T12 and less than 2 days for T15, for 600 epochs. For hyperparameter tuning, we use 16 NVIDIA A40 GPUs for phoneme decoding, completing the process in less than 2 days. Checkpoint selection follows the same procedure as for transformer-based models.

**LLM Decoder.** For all decoders, we train using the AdamW optimizer with *bfloat16* precision for 150 epochs. For transformer encoders, sentence-level decoding requires less than 1 day for T12 and less than 2 days for T15 on a single NVIDIA A40 or A100 GPU. For RNN, this process takes less than 1.5 days for T12 and less than 4 days for T15 on a single NVIDIA A40 or A100 GPU. Small-sized LLMs, including 0.6B, 1.5B, and 1.7B models, are trained on a single NVIDIA A40 or L40 GPU (48 GB memory). Larger LLMs around 7B parameters are trained on two NVIDIA A40 or L40 GPUs with *DeepSpeed ZeRO-3* optimization. In all cases, checkpoint selection is based on the validation WER.

## S  THE USE OF LARGE LANGUAGE MODELS (LLMS)

We used LLMs to polish the writing of this paper. Specifically, LLMs were applied to improve grammar and clarity. No scientific contribution or results was generated by an LLM, and all AI-assisted edits were verified by the researchers.

