# OpenReview forum: "A cross-species neural foundation model for end-to-end speech decoding"
_ICLR.cc/2026/Conference — ICLR 2026 Poster_

### Official Review · Reviewer_6BdX · 2025-10-17

**Soundness:** 2
**Presentation:** 2
**Contribution:** 1
**Rating:** 2
**Confidence:** 4

**Summary:**

This work introduces BIT, an end-to-end brain-to-text speech decoding framework that translates intracortical neural activity into text by combining a transformer-based, cross-task/cross-species pretrained neural encoder with an audio large language model (LLM) decoder. The authors design an architecture that supports both cascaded and end-to-end decoding by leveraging self-supervised masked modeling pretraining on extensive human and monkey datasets, phoneme-level finetuning, and contrastive learning for cross-modal neural-text alignment. The method is evaluated on the Brain-to-Text Benchmark '24 and '25 for both attempted and imagined speech.

**Strengths:**

1. The manuscript is well written and easy to follow. Numerous tables and figures provide detailed results, ablations, and interpretability analysis, helping readers understand model choices and results. The supplemental materials provide many clarifying details and results.

2. Based on the well-tuned architecture in Time-Masked Transformer [7], MAE-based pre-training further improves `~0.5%` in two speech decoding datasets [2,8] in Table 1.

3. Explore the effects of audio LLMs for end-to-end speech decoding, instead of LLMs used in [9].

**Weaknesses:**

1. The significant performance drop in the end-to-end model (Fig. 2A) casts doubt on the effectiveness of the proposed Audio-LLM Decoder (Fig. 1C). This finding is consistent with prior work [4].

2. The pre-training approach lacks neuroscientific justification. It aggregates intracranial data from different brain regions (e.g., motor [1] vs. speech production [2]) and species, without explaining how such disparate neural computations can share a common latent representation. Perhaps applying more data augmentation (e.g., timing jittering) during pre-training and using a single subject’s data would be enough to compensate for the reported performance differences (Figure 2). Besides, the motivation for removing time masking during fine-tuning is unclear (Line 223), which is a core module in Time Masked Transformer [7].

3. Using audio models for supervision is not novel [3], where the results indicate that audio latent guidance is less effective than sequential phoneme labels under phoneme error rate (PER) evaluation.

4. The use of Masked Autoencoder (MAE) pre-training is a well-established technique and does not represent a significant methodological novelty [5,6].

5. The proposed BIT model appears to have a significantly higher training cost than the Time-Masked Transformer baseline. To better contextualize this, the authors should include a runtime comparison, detailing the reported 260-hour pre-training time and the computational overhead of the Audio-LLM Decoder.

**References**:

[1] Willett F R, Avansino D T, Hochberg L R, et al. High-performance brain-to-text communication via handwriting[J]. Nature, 2021, 593(7858): 249-254.

[2] Willett F R, Kunz E M, Fan C, et al. A high-performance speech neuroprosthesis[J]. Nature, 2023, 620(7976): 1031-1036.

[3] Metzger S L, Littlejohn K T, Silva A B, et al. A high-performance neuroprosthesis for speech decoding and avatar control[J]. Nature, 2023, 620(7976): 1037-1046.

[4] Jiang W B, Wang Y, Lu B L, et al. NeuroLM: A universal multi-task foundation model for bridging the gap between language and EEG signals[J]. arXiv preprint arXiv:2409.00101, 2024.

[5] Ye J, Collinger J, Wehbe L, et al. Neural data transformer 2: multi-context pretraining for neural spiking activity[J]. Advances in Neural Information Processing Systems, 2023, 36: 80352-80374.

[6] Kapoor J, Schulz A, Vetter J, et al. Latent diffusion for neural spiking data[J]. Advances in Neural Information Processing Systems, 2024, 37: 118119-118154.

[7] Feghhi E, Kaasyap S, Hadidi N, et al. Time-Masked Transformers with Lightweight Test-Time Adaptation for Neural Speech Decoding[J]. arXiv preprint arXiv:2507.02800, 2025.

[8] Card N S, Wairagkar M, Iacobacci C, et al. An accurate and rapidly calibrating speech neuroprosthesis[J]. New England Journal of Medicine, 2024, 391(7): 609-618.

[9] Feng S, Liu H, Wang Y, et al. Towards an end-to-end framework for invasive brain signal decoding with large language models[J]. arXiv preprint arXiv:2406.11568, 2024.

**Questions:**

See concerns.

---

> ### Author Response · Authors · 2025-11-21
> **Response to Reviewer 6BdX**
>
> *Strengths*:
> > Based on the well-tuned architecture in Time-Masked Transformer [7], MAE-based pre-training further improves ~0.5% in two speech decoding datasets [2,8] in Table 1.
>
> Thank you for your comment on our strengths. However, **our improvement is larger than the reviewer’s claim**. In Table 1, the proper comparison should be made between the base models without ensembling. In this setting, our encoder reduces the WER by ~1.6% compared to Time-Masked Transformer. With ensembling, the gain is ~0.5% WER, although this depends on ensemble size. Moreover, **we rank #1 on two well-recognized and highly competitive brain-to-text benchmarks, confirming our encoder outperforms all existing ones in this field.**
>
> *Weaknesses*:
> > 1. The proposed BIT model appears to have a significantly higher training cost than the Time-Masked Transformer baseline. To better contextualize this, the authors should include a runtime comparison, detailing the reported 260-hour pre-training time and the computational overhead of the Audio-LLM Decoder.
>
> **The reviewer’s statement is factually incorrect. We did not report a 260-hour pretraining time; the actual time is 48 hours** (Appendix O, “Training Details”).
>
> **Our goal is to reduce**, not increase, **computational burden for the BCI community** by providing an open-source pretrained model that others can easily reuse. Such models transfer well to new datasets/tasks with minimal fine-tuning, which will benefit researchers with limited data or compute. Although pretraining takes 48 hours on a single GPU, **fine-tuning requires a similar amount of time as the Time-Masked Transformer.**
>
> **Runtime**: Although the n-gram LM is ready to use once trained, its initial training is also compute-intensive. Fine-tuning the audio-LLM is fast (<1 day), and inference speed is comparable: <1 s for end-to-end versus ~0.25 s for cascaded models: a negligible difference in practice. Moreover, the **end-to-end system can be further accelerated** with hybrid architectures (e.g., SSM modules) and modern GPUs, **whereas cascaded models remain CPU-bound.**
>
> > 2. The significant performance drop in the end-to-end model (Fig. 2A) casts doubt on the effectiveness of the proposed Audio-LLM Decoder (Fig. 1C). This finding is consistent with prior work [4].
>
> We appreciate the opportunity to address this concern. Adopting an end-to-end model is necessary because:
>
> (1) Although end-to-end models currently underperform cascaded ones, they are crucial for future brain-to-text models. Cascaded models excel in the small-data regime because the 5-gram LM provides a strong inductive bias that compensates for limited data. End-to-end models remove this bias and learn more flexible neural-to-text mappings; while such biases help with scarce data, they become limiting as datasets grow. Similar patterns are seen in NLP and vision, where **models with inductive biases perform better with small data but flexible and expressive architectures dominate at scale**. Developing end-to-end models now will prepare us to fully exploit future large-scale datasets.
>
> (2) **End-to-end models will become especially powerful as recordings expand to cortical regions that encode higher-level information beyond phonemes.** In these areas, neural activity often reflects semantic, syntactic, or prosodic structure rather than low-level articulatory features. By directly connecting neural and LLM representations, end-to-end models preserve this higher-order structure, whereas cascaded models lose much of it by forcing the data through phoneme-based intermediates.
>
> (3) **The inference strategies of the cascaded and end-to-end models are not directly comparable**. The cascaded model uses a 5-gram LM with beam search and rescoring, while the end-to-end model uses simple greedy decoding without beam search. To ensure fair comparisons, we removed beam search and rescoring, which reduced the performance gap for attempted speech to below ~5% WER (Table 1). In the paper, we kept the beam search for the cascaded model, since we want to compare against its strongest baseline rather than weaken its performance in favor of us.
>
> ### Table 1: Attempted speech decoding (T12)
> | Model                             | WER (%) |
> |----------------------------------|---------|
> | Cascaded (with beam search)      | 6.4%    |
> | Cascaded (no beam search)        | 10.34%  |
> | End-to-end                       | 15.7%   |
>
> (4) **Cascaded models are not ideal in all scenarios**: when the phoneme decoder is suboptimal, the 5-gram LM in the cascaded system struggles to generate accurate sentences. For example, Figure 2B shows that the cascaded and end-to-end models perform comparably in imagined speech decoding.

---

> ### Author Response · Authors · 2025-11-21
> **Response to Reviewer 6BdX (continued)**
>
> > 3. The pre-training approach lacks neuroscientific justification. It aggregates intracranial data from different brain regions (e.g., motor [1] vs. speech production [2]) and species, without explaining how such disparate neural computations can share a common latent representation. Perhaps applying more data augmentation (e.g., timing jittering) during pre-training and using a single subject’s data would be enough to compensate for the reported performance differences (Figure 2). Besides, the motivation for removing time masking during fine-tuning is unclear (Line 223), which is a core module in Time Masked Transformer [7].
>
> The justification for cross-species SSL pretraining is twofold: 1) to learn stable Utah-array representations that capture spiking patterns and probe-specific structure robust to noise; 2) to build a strong neural encoder for multimodal alignment with LLMs, as encoders pretrained on more data align more effectively. The shared latent representation centers on Utah-array probes, capturing spiking and probe-specific structure that enables the model to overcome sensor variability across participants, species, and tasks.
>
> Data augmentation: we already apply extensive augmentation (Gaussian noise and time masking on half the neural data per trial) when training the baseline transformer from scratch. As shown in Figure 2, the pretrained encoder still outperforms this baseline and ranks #1 on both brain-to-text benchmarks. This underscores the value of large-scale pretraining.
>
> Masking: time masking is disabled only during fine-tuning (not when training baseline transformers from scratch), as it slightly hurts decoding performance. The pretrained encoder has already learned robustness to noise and variable spiking patterns through extensive augmentation during SSL pretraining.
>
> > 4. Using audio models for supervision is not novel [3], where the results indicate that audio latent guidance is less effective than sequential phoneme labels under phoneme error rate (PER) evaluation.
>
> There seem to be some misunderstandings here. To avoid confusion for the AC and reviewers, we clarify:
>
> (1) The paper cited by the reviewer uses an audio model as a vocoder for brain-to-voice decoding, whereas we focus on brain-to-text decoding and are the first to connect a neural encoder to an audio-LLM. Our results (Figure 3) show that **audio latent guidance is more effective than semantic latent guidance** from text-only LLMs.
>
> (2) The current underperformance of the end-to-end model relative to the cascaded system is not due to ineffective audio guidance but to the challenge of fine-tuning a large LLM with limited human data. In fact, audio-LLMs provide stronger guidance than text-only LLMs (Figure 3). For an explanation of why end-to-end models remain important despite limited data, please see our response to Weaknesses #2.
>
> > 5. The use of Masked Autoencoder (MAE) pre-training is a well-established technique and does not represent a significant methodological novelty [5,6].
>
> We appreciate the time the reviewer took to review our paper, but this comment here is potentially misleading. We never claimed masked-modeling pretraining as our methodological novelty. Our contribution is demonstrating that cross-species, cross-task SSL pretraining yields consistent gains for attempted speech decoding and larger gains for imagined speech. This is an important contribution to the intracortical BCI field.
>
> Similarly, the Time-Masked Transformer [1] recommended by the reviewer also used MAE-style masking and was recognized for showing that time masking is an effective augmentation against overfitting. **Our novelty** likewise lies not in the MAE objective itself, but in **demonstrating that large-scale SSL pretraining across species and tasks improves brain-to-text decoding by addressing Utah-array sensor variability.**
>
> [1] Feghhi E, Kaasyap S, Hadidi N, et al. Time-Masked Transformers with Lightweight Test-Time Adaptation for Neural Speech Decoding[J]. arXiv preprint arXiv:2507.02800, 2025.

---

> ### Comment · Reviewer_6BdX · 2025-11-25
> **Official Comment by Reviewer 6BdX**
>
> Thanks to the author for their feedback on the review comments.
>
> **W1**:
>
> > ...Cascaded models excel in the small-data regime because the 5-gram LM provides a strong inductive bias that compensates for limited data...Developing end-to-end models now will prepare us to fully exploit future large-scale datasets.
>
> Thank you for clarifying the potential of BIT. However, the Brain-to-Text Benchmark is currently the largest publicly available microelectrode arrays dataset for speech decoding, especially the Brain-to-Text Benchmark '25 dataset, which spans nearly two years. **The results on these two datasets are currently insufficient to support this conclusion.** Besides, to the best of my knowledge, although Card et al. did not release the entire dataset [1], training a Transformer variant (Roformer [2]) on the entire dataset (`595` hours of task recordings) achieved a word error rate of less than `1%`.
>
> > ...In these areas, neural activity often reflects semantic, syntactic, or prosodic structure rather than low-level articulatory features...
>
> Thank you for clarifying the potential of BIT. To the best of my knowledge, there is no public dataset to evaluate such a hypothesis. In the Brain-to-Text Benchmark, the microelectrode array was primarily implanted in the vSMC, an area mainly responsible for speech motor functions, which corresponds to phoneme labeling. Furthermore, the two participants primarily performed attempted speech attempts and imagery speech tasks, in which the vSMC is responsible for controlling facial muscles to produce speech. **To demonstrate the model's potential, the best approach would be to collect a relevant dataset and then conduct an evaluation.**
>
> > ...The cascaded model uses a 5-gram LM with beam search and rescoring, while the end-to-end model uses simple greedy decoding without beam search...For example, Figure 2B shows that the cascaded and end-to-end models perform comparably in imagined speech decoding.
>
> Thank you for providing the additional results. I acknowledge the great efforts made in this exploratory work. **If the model achieves better performance even with increased training overhead, it would be a strong indication of its superiority.**
>
> **References**:
>
> [1] Card N S, Wairagkar M, Iacobacci C, et al. An accurate and rapidly calibrating speech neuroprosthesis[J]. New England Journal of Medicine, 2024, 391(7): 609-618.
>
> [2] Su J, Ahmed M, Lu Y, et al. Roformer: Enhanced transformer with rotary position embedding[J]. Neurocomputing, 2024, 568: 127063.
>
> **W2**:
>
> > The justification for cross-species SSL pretraining is twofold...
>
> Thank you for clarifying the potential of cross-species SSL. **For the novelty of cross-species and cross-subject SSL of neural spike recordings, please refer to [1,2].** For example, NDT3 has already used `2k` hours of data for pre-training.
>
> > Data augmentation...The pretrained encoder has already learned robustness to noise and variable spiking patterns through extensive augmentation during SSL pretraining.
>
> Thank you for clarifying the advanced results of SSL pretraining. I acknowledge that MAE-based pretraining is indeed a well-established technique.
>
> **References**:
>
> [1] Ye J, Collinger J, Wehbe L, et al. Neural data transformer 2: multi-context pretraining for neural spiking activity[J]. Advances in Neural Information Processing Systems, 2023, 36: 80352-80374.
>
> [2] Ye J, Rizzoglio F, Smoulder A, et al. A Generalist Intracortical Motor Decoder[J]. Advances in Neural Information Processing Systems, 2025.
>
> **W3**:
>
> Thank you for clarifying the misunderstanding. Please refer to **W1**.
>
> **W4**:
>
> Thank you for clarifying the core contribution. Limited performance improvements and constrained innovation limit this core statement. Please refer to **W1**.
>
> **W5**:
>
> Thank you for clarifying the misunderstanding. My intention was to determine the time required for pre-training on `269+98` hours of data. I apologize if this has caused any misunderstanding. For concerns regarding core contributions to the work, please refer to **W1**.

---

> ### Author Response · Authors · 2025-11-25
> **Follow-up Response to Reviewer 6BdX**
>
> We thank the reviewer for their comments.
>
> > W1: ... However, the Brain-to-Text Benchmark is currently the largest publicly available microelectrode arrays dataset for speech decoding, especially the Brain-to-Text Benchmark '25 dataset, which spans nearly two years. The results on these two datasets are currently insufficient to support this conclusion. Besides, to the best of my knowledge, although Card et al. did not release the entire dataset [1], training a Transformer variant (Roformer [2]) on the entire dataset (595 hours of task recordings) achieved a word error rate of less than 1%.
>
> We would like to clarify a **factual inaccuracy** in the statement. Although the Brain-to-Text Benchmark is currently the largest public intracortical dataset, it contains **only 10,948 labeled sentences**, which is far smaller than what is typically used in NLP or CV. Thus, the intracortical human BCI field remains a **small-data domain**. **The claim that the dataset "spans nearly 2 years" is technically correct but misleading, as only the labeled portions of those recordings (the 10,948 sentences) are public.**
>
> Our results on the 2 datasets already show that pretraining the end-to-end model on more data improves attempted-speech decoding. As shown in Table 1 of the paper, our proposed end-to-end model, pretrained on extensive Utah-array data, reduces WER by nearly 50% compared to the previous end-to-end method (24.69% → 10.22%). This provides **strong evidence that additional data leads to better end-to-end decoding performance.**
>
> To fairly compare with the model from Card et al., as the reviewer suggests, both cascaded and end-to-end models would need access to the same pretraining data. However, those data are not publicly available.
>
> > ... To the best of my knowledge, there is no public dataset to evaluate such a hypothesis ... To demonstrate the model's potential, the best approach would be to collect a relevant dataset and then conduct an evaluation.
>
> This is precisely why we developed an end-to-end model: once researchers collect such datasets from higher-level brain areas, they can build on our mature decoding framework rather than developing a new end-to-end method from scratch.
>
> > Thank you for providing the additional results. I acknowledge the great efforts made in this exploratory work. If the model achieves better performance even with increased training overhead, it would be a strong indication of its superiority.
>
> **The statement is misleading and downplays the contribution of our work.** **Our pretrained neural encoder is** not "exploratory" but **state-of-the-art**, as demonstrated by achieving 1st place on both Brain-to-Text benchmarks. Our end-to-end approach is also innovative in offering a forward-looking framework for brain-to-text decoding. **Focusing only on the new end-to-end component while ignoring our large-scale pretraining contribution, which enables the SOTA results, risks misinforming the AC and other reviewers.**
>
> > W2: Thank you for clarifying the potential of cross-species SSL. For the novelty of cross-species and cross-subject SSL of neural spike recordings, please refer to [1,2]. For example, NDT3 has already used 2k hours of data for pre-training.
>
> Although prior intracortical BCI studies (POSSAM [1], NDT3 [2]) used monkey data to mitigate small-data issues, POSSAM did not transfer to speech decoding, and NDT3 achieved only limited cross-species generalization. In contrast, we show that large-scale pretraining across species and tasks provides clear transfer benefits, achieving SOTA human speech decoding and ranking #1 on two well-recognized and highly competitive brain-to-text benchmarks.
>
> > W4: Thank you for clarifying the core contribution. Limited performance improvements and constrained innovation limit this core statement. Please refer to W1.
>
> Again, this statement is misleading. The **performance gains** from our proposed methods **are substantial**: **large-scale pretraining of the neural encoder achieves SOTA (1st place)** on both Brain-to-Text benchmarks, and **our end-to-end approach improves the previous end-to-end baseline performance by 50%**. Given these strong empirical improvements, it is unclear why the reviewer characterizes our contributions as offering only “limited performance improvement.”
>
> Re: "constrained innovation" - Please see our response on the significance of cross-species transfer (reply to W2 in this thread) and the importance of the end-to-end approach (reply to Weakness #2 in our initial "Response to Reviewer 6BdX").
>
> [1] Hee-Woon Ryoo, Avery, et al. "Generalizable, real-time neural decoding with hybrid state-space models. NeurIPS 2025.
>
> [2] Ye, Joel, et al. "A Generalist Intracortical Motor Decoder." bioRxiv 2025.

---

### Official Review · Reviewer_1Ni1 · 2025-10-31

**Soundness:** 2
**Presentation:** 2
**Contribution:** 2
**Rating:** 4
**Confidence:** 2

**Summary:**

This paper investigates the use of self-supervised learning (SSL) combined with Transformer architectures for decoding inner speech from EEG signals. The authors pre-train an SSL model on a large-scale EEG dataset and evaluate it on several downstream decoding tasks. The work aims to bridge the gap between recent advances in representation learning and neural decoding applications.

**Strengths:**

- The paper addresses an important and technically challenging problem: decoding inner speech from EEG, a task that remains largely open and scientifically valuable.
- The experimental design is systematic and the paper provides clear descriptions of the model, datasets, and training procedures.
- Empirical results show consistent improvements over baseline methods, indicating that the approach is effective in leveraging large-scale pre-training for EEG representation learning.

**Weaknesses:**

1. **Potential Data Leakage in Pre-training**
   The SSL model was pre-trained on a large EEG dataset that includes data from some subjects who also appear in the test set. Although SSL does not rely on explicit labels, this overlap means that data from the test set contributed to the model’s representational space. As a result, the reported improvements might partially reflect familiarity with certain subjects rather than genuine generalization to unseen data. Clarifying or mitigating this issue would be important for ensuring the validity of the conclusions.

2. **Characterization of Model Novelty**
   The Introduction suggests that prior work "does not explore modern architectures, such as Transformer." This statement may be somewhat overstated. The Transformer architecture has been extensively studied across many fields, including neural data modeling. The novelty here lies more in its application to *inner speech decoding* rather than in the architecture itself. The framing could be refined to more accurately reflect this distinction.

3. **Limited Theoretical Innovation**
   The paper primarily combines existing components—SSL pre-training, Transformer encoders, and standard EEG decoding techniques—on publicly available data. While the integration is well executed and empirically valuable, the theoretical innovation appears limited. The contribution may therefore be more in engineering practice than in advancing fundamental understanding in machine learning or neuroscience.

**Questions:**

Since the contribution is primarily empirical, it would be helpful to better understand the theoretical or conceptual motivation behind combining SSL and Transformer.   For instance, what properties of EEG data make Transformer architectures particularly suitable?

---

> ### Author Response · Authors · 2025-11-21
> **Response to Reviewer 1Ni1**
>
> We thank the reviewer for recognizing the importance of our work and acknowledging the rigor of our experiments and strong empirical results. We address the questions point-by-point below.
>
> *Weaknesses*:
>
> > 1. The SSL model was pre-trained on a large EEG dataset that includes data from some subjects who also appear in the test set. Although SSL does not rely on explicit labels, this overlap means that data from the test set contributed to the model’s representational space. As a result, the reported improvements might partially reflect familiarity with certain subjects rather than genuine generalization to unseen data. Clarifying or mitigating this issue would be important for ensuring the validity of the conclusions.
>
> Thank you for raising this point. First, we would like to clarify that we used intracortical spiking data from Utah-array probes, not EEG. The pretraining procedure does not cause data leakage since it is entirely unlabeled. In many fields, including or excluding fine-tuning datasets during pretraining depends on the type of generalization being tested: **excluding them tests generalization to unseen data, while including them tests generalization to unseen tasks.** Here, we are interested in improving transfer to speech decoding tasks, and therefore chose to include the human speech data during pretraining. Moreover, we aimed to use as much human data as possible to stabilize the learned representations, given the scarcity of human data.To address this concern directly, we also repeated pretraining without human speech datasets to test generalization to unseen data. We observe no substantial difference in attempted speech decoding performance regardless of whether the human finetuning data are included during pretraining. The full results are provided in Appendix Section P and Table 8.
>
> > 2. The Introduction suggests that prior work "does not explore modern architectures, such as Transformer." This statement may be somewhat overstated. The Transformer architecture has been extensively studied across many fields, including neural data modeling. The novelty here lies more in its application to inner speech decoding rather than in the architecture itself. The framing could be refined to more accurately reflect this distinction.
>
> This statement refers to prior intracortical BCI work using Utah-array data, which is why only those studies were cited. Transformer architectures remain largely unexplored in this field, where previous state-of-the-art models are mostly RNN-based.
>
> > 3. The paper primarily combines existing components—SSL pre-training, Transformer encoders, and standard EEG decoding techniques—on publicly available data. While the integration is well executed and empirically valuable, the theoretical innovation appears limited. The contribution may therefore be more in engineering practice than in advancing fundamental understanding in machine learning or neuroscience.
>
> We appreciate the reviewer’s question and will update the preprint to highlight our contributions to both ML and neuroscience:
>
> (1) **Neuroscience and BCI**: This work provides contributions that advance fundamental understanding in neuroscience and intracortical BCI. While prior intracortical studies such as POSSAM [1] and NDT3 [2] attempted cross-species transfer to improve human BCIs, POSSAM did not transfer to speech decoding, and NDT3 achieved limited cross-species generalization. In contrast, **we are the first to demonstrate that large-scale pretraining across species and tasks provides clear transfer gains, achieving state-of-the-art human speech decoding and ranking #1 on two well-recognized and highly competitive brain-to-text benchmarks.**
>
> (2) **ML**: To the best of our knowledge, we are one of only two works that have successfully connected intracortical neural activity with LLMs, and we have achieved the best modality alignment to date. This connection enables direct comparison between neural and language/speech representations, offering a new avenue to study how LLMs encode speech and language structures in relation to the brain.
>
> [1] Hee-Woon Ryoo, Avery, et al. "Generalizable, real-time neural decoding with hybrid state-space models. NeurIPS 2025.
>
> [2] Ye, Joel, et al. "A Generalist Intracortical Motor Decoder." bioRxiv 2025.

---

> > ### Author Response · Authors · 2025-11-21
> > **Response to Reviewer 1Ni1 (continued)**
> >
> > *Questions*:
> > > Since the contribution is primarily empirical, it would be helpful to better understand the theoretical or conceptual motivation behind combining SSL and Transformer. For instance, what properties of EEG data make Transformer architectures particularly suitable?
> >
> > First, we would like to clarify that we used intracortical spiking data from Utah-array probes, not EEG. We chose the transformer because it offers a scalable architecture for sequence data like spiking activity:
> >
> > (1) Transformers capture long-range temporal dependency, and we use time-patch tokenization to further capture short-range dependency.
> >
> > (2) Transformers are more computationally efficient than RNNs, making them better suited for real-time BCI use.
> > We use SSL because it is label-free and allows pretraining on additional Utah-array spiking data beyond human speech datasets. Including more data helps the model learn stable, noise-robust representations, which is especially valuable for human BCI applications with limited data.

---

### Official Review · Reviewer_pds5 · 2025-11-01

**Soundness:** 2
**Presentation:** 3
**Contribution:** 2
**Rating:** 4
**Confidence:** 3

**Summary:**

This paper presents BIT, an end-to-end framework for brain-to-text speech decoding aiming to translate neural activity directly into coherent sentences. The core contribution include a transformer-based neural encoder pretrained on large-scale, cross-task and cross-species neural datasets, integration with Audio-LLM. The proposed method achieved a lower word error rate (WER) on the test set compared to the baseline.

**Strengths:**

S1: This paper builds an end-to-end architecture, integrating neural signals into a multimodal LLM and performing modality alignment.
S2: This paper conducts additional analyses, such as ablation studies across different models and comparisons of the representational geometry of neural embeddings against that of LLMs.
S3: This paper provides a relatively thorough discussion of its limitations.

**Weaknesses:**

W1: The evaluation metrics only involve WER, which seems insufficiently comprehensive.  Although the paper introduces finetuning for sentence-level decoding, it evaluates performance solely at the word level. Incorporating sentence-level evaluation metrics, such as semantic similarity, would likely provide a more complete assessment. In addition, a more detailed analysis of the error words should be provided. For example, does the same word sometimes appear correctly and sometimes incorrectly? Are common words recognized more accurately? Are words that appear infrequently in the training data more prone to errors?
W2: The paper introduces cross-species (monkey) data for training. Although this approach is relatively novel, the paper lacks further discussion on how the monkey data affect the training performance. I am also interested in understanding the impact of different proportions of monkey and human data on the results.
W3: The end-to-end architecture proposed in the paper appears to underperform cascade systems in terms of both effectiveness and efficiency in practical applications, which raises doubts about the necessity of adopting an end-to-end framework for the brain-to-text task. In particular, given the limited amount of data available in this domain, it is questionable whether an LLM-based end-to-end architecture can be effectively trained.

**Questions:**

Q1: In the BIT loss, the cross-entropy loss and the contrastive loss are simply added together. What would be the effect if the weighting between these two loss terms were adjusted?
Q2: The training details in Appendix O show that hundreds of epochs are used for training the different components. Is there any theoretical justification for such a large number of epochs, and does this raise concerns about potential overfitting?
Q3: The motivation for using an Audio-LLM requires further explanation. Why not adopt a VLM or Omni model instead?
Q4: The reason for adopting LoRA is not entirely clear. Would full-parameter training yield better performance?
Q5: The concepts of attempted speech and imagined speech are mentioned multiple times early in the paper, but they are not explained until Section 4.1. It would improve readability to introduce and clarify these concepts earlier in the text.

---

> ### Author Response · Authors · 2025-11-20
> **Response to Reviewer pds5**
>
> > W1: The evaluation metrics only involve WER, which seems insufficiently comprehensive. Although the paper introduces finetuning for sentence-level decoding, it evaluates performance solely at the word level. Incorporating sentence-level evaluation metrics, such as semantic similarity, would likely provide a more complete assessment. In addition, a more detailed analysis of the error words should be provided. For example, does the same word sometimes appear correctly and sometimes incorrectly? Are common words recognized more accurately? Are words that appear infrequently in the training data more prone to errors?
>
> Thank you for the comment. Sentence-level metrics can be useful in some BCI settings, but they are mainly applied to non-invasive BCIs with low accuracy (high WER), where word-level evaluation is unreliable. In our case, the invasive recordings provide high-fidelity neural signals that lead to extremely low WER (>97% accuracy on participant T15), making word-level accuracy both meaningful and sufficient. Sentence-level metrics would thus add little beyond WER, though we agree that a systematic analysis of word and phoneme errors is valuable. To address this, we performed both phoneme-level and word-level error analyses to better characterize decoder behavior. Overall, the phoneme errors follow known phonetic similarity patterns and match expected decoder behavior, with only a small number of word-level errors that mostly involve near-neighbor confusions. Details can be found in Appendix Section N and Figures 6–7.
>
> > W2: The paper introduces cross-species (monkey) data for training. Although this approach is relatively novel, the paper lacks further discussion on how the monkey data affect the training performance. I am also interested in understanding the impact of different proportions of monkey and human data on the results.
>
> We agree that understanding how performance scales with varying proportions of human and monkey data is important. A natural way to show this is through a scaling curve showing decoding performance versus pretraining data size. To address this question, we performed scaling experiments for both human-only and cross-species pretraining, examining how gradually increasing the amount of human versus monkey data influences attempted speech decoding performance for participant T15. Across both cascaded and end-to-end models, we find that adding human data yields larger performance gains than augmenting the model with monkey data. The full results are presented in Appendix Section O and Figure 8.

---

> > ### Author Response · Authors · 2025-11-21
> > **Response to Reviewer pds5 (continued)**
> >
> > > W3: The end-to-end architecture proposed in the paper appears to underperform cascade systems in terms of both effectiveness and efficiency in practical applications, which raises doubts about the necessity of adopting an end-to-end framework for the brain-to-text task. In particular, given the limited amount of data available in this domain, it is questionable whether an LLM-based end-to-end architecture can be effectively trained.
> >
> > We appreciate the opportunity to address this concern. Adopting an end-to-end model is necessary because:
> >
> > (1) Although end-to-end models currently underperform cascaded ones, they are crucial for future brain-to-text models. Cascaded models excel in the small-data regime because the 5-gram LM provides a strong inductive bias that compensates for limited data. End-to-end models remove this bias and learn more flexible neural-to-text mappings; while such biases help with scarce data, they become limiting as datasets grow. Similar patterns are seen in NLP and vision, where **models with inductive biases perform better with small data but flexible and expressive architectures dominate at scale.** Developing end-to-end models now will prepare us to fully exploit future large-scale datasets.
> >
> > (2) **End-to-end models will become especially powerful as recordings expand to cortical regions that encode higher-level information beyond phonemes.** In these areas, neural activity often reflects semantic, syntactic, or prosodic structure rather than low-level articulatory features. By directly connecting neural and LLM representations, end-to-end models preserve this higher-order structure, whereas cascaded models lose much of it by forcing the data through phoneme-based intermediates.
> >
> > (3) **The inference strategies of the cascaded and end-to-end models are not directly comparable.** The cascaded model uses a 5-gram LM with beam search and rescoring, while the end-to-end model uses simple greedy decoding without beam search. To ensure fair comparisons, we removed beam search and rescoring, which reduced the performance gap for attempted speech to below ~5% WER (Table 1). In the paper, we kept the beam search for the cascaded model, since we want to compare against its strongest baseline rather than weaken its performance in favor of us.
> >
> > ### Table 1: Attempted speech decoding (T12)
> > | Model                             | WER (%) |
> > |----------------------------------|---------|
> > | Cascaded (with beam search)      | 6.4%    |
> > | Cascaded (no beam search)        | 10.3%  |
> > | End-to-end                       | 15.7%   |
> >
> > (4) **Cascaded models are not ideal in all scenarios**: when the phoneme decoder is suboptimal, the 5-gram LM in the cascaded system struggles to generate accurate sentences. For example, Figure 2B shows that the cascaded and end-to-end models perform comparably in imagined speech decoding.
> >
> > **Efficiency**: the current **end-to-end system** is not yet optimized for real-time use but **can be improved** with hybrid architectures (e.g., SSM-based modules) and modern GPU advances, **whereas cascaded models remain CPU-bound.**
> >
> > > Q1: In the BIT loss, the cross-entropy loss and the contrastive loss are simply added together. What would be the effect if the weighting between these two loss terms were adjusted?
> >
> > Thank you for the question. This experiment is in progress, and we will share results soon.
> >
> > > Q2: The training details in Appendix O show that hundreds of epochs are used for training the different components. Is there any theoretical justification for such a large number of epochs, and does this raise concerns about potential overfitting?
> >
> > During pretraining and transformer training from scratch, we apply time masking for data augmentation. Training over many epochs exposes the model to diverse augmented samples, which helps prevent overfitting. During phoneme-level fine-tuning, we also inject Gaussian noise as data augmentation, which requires many training steps to build robustness to noise. For LLM fine-tuning, masking is disabled, but we still train for many epochs because neural datasets are much smaller than typical LLM datasets, where each epoch may contain millions of steps, and we also use smaller batch sizes. Therefore, more epochs are needed for the model to learn effectively. Finally, overfitting is minimal, as all model checkpoints for evaluation are selected based on validation performance.

---

> > > ### Author Response · Authors · 2025-11-21
> > > **Response to Reviewer pds5 (continued)**
> > >
> > > > Q3: The motivation for using an Audio-LLM requires further explanation. Why not adopt a VLM or Omni model instead?
> > >
> > > Thank you for the question. The rationale is explained in the paper (L191–196; L366–370) and summarized here. We prioritize audio LLMs over text-only LLMs because:
> > >
> > > (1) **Although this is a brain-to-text task, the participant performed a speech task.** Neural activity was recorded from motor and speech areas that encode phoneme-level structure [1]. Audio-LLMs, pretrained on speech recognition tasks, naturally capture phonetic and prosodic patterns [2].
> > >
> > > (2) **Neural signals, like speech waveforms, are continuous time-series data.** Speech models thus transfer better to neural activity, whereas text-only LLMs lack the inductive biases needed for temporal signal modeling [3].
> > > Finally, the choice of LLM decoder depends on the task: a visual task would favor a VLM, while multimodal tasks (e.g., audiovisual stimuli) would benefit from an Omni model.
> > >
> > > > Q4: The reason for adopting LoRA is not entirely clear. Would full-parameter training yield better performance?
> > >
> > > We use LoRA because the LLM is too large for full fine-tuning, which would require far more memory and compute. LoRA enables efficient adaptation with manageable computational cost.
> > >
> > > > The concepts of attempted speech and imagined speech are mentioned multiple times early in the paper, but they are not explained until Section 4.1. It would improve readability to introduce and clarify these concepts earlier in the text.
> > >
> > > We agree that introducing these concepts earlier would improve readability and will clarify them at first mention in the main text.
> > >
> > > [1] Willett, Francis R., et al. "A high-performance speech neuroprosthesis." Nature 2023.
> > >
> > > [2] Chu, Yunfei, et al. "Qwen2-audio technical report." arXiv preprint 2024.
> > >
> > > [3] Millet, Juliette, et al. "Toward a realistic model of speech processing in the brain with self-supervised learning." NeurIPS 2022.

---

> > > > ### Author Response · Authors · 2025-11-25
> > > > **Update on Q1 from Reviewer pds5**
> > > >
> > > > > Q1: In the BIT loss, the cross-entropy loss and the contrastive loss are simply added together. What would be the effect if the weighting between these two loss terms were adjusted?
> > > >
> > > > Thank you for the question. We conducted an ablation study to investigate how changing the weight ratio between the LLM’s cross-entropy loss and the InfoNCE loss for contrastive-learning–based modality alignment affects performance. The total weight sums to 1, but the relative ratio between the two losses varies. Using data from participant T15, we examined how different weight ratios influence attempted-speech decoding. As shown in Table 2, maintaining the original ratio yields the best decoding performance, while more aggressive weight ratios lead to notably worse performance.
> > > >
> > > > **Table 2: Weight Ratio (Cross Entropy Loss : InfoNCE)**
> > > >
> > > > | Weight Ratio | WER   |
> > > > |--------------|-------|
> > > > | 1 : 1        | 11.06 |
> > > > | 2 : 1        | 12.79 |
> > > > | 4 : 1        | 13.62 |
> > > > | 1 : 2        | 13.78 |
> > > > | 1 : 4        | 15.36 |

---

### Official Review · Reviewer_GRt5 · 2025-11-03

**Soundness:** 3
**Presentation:** 3
**Contribution:** 2
**Rating:** 4
**Confidence:** 3

**Summary:**

Advancing speech brain-computer interfaces (BCIs) to restore communication for individuals with paralysis, this paper (under review at ICLR 2026, titled *Decoding Inner Speech with an End-to-End Brain-to-Text Neural Interface*) addresses the key limitation of existing cascaded frameworks—where recurrent neural networks (RNNs) decode phonemes first before assembling sentences with n-gram language models (LMs), preventing end-to-end optimization and often causing mismatches between phoneme error rate (PER) and word error rate (WER)—by proposing the end-to-end BraIn-to-Text (BIT) framework, whose core is a transformer neural encoder pretrained via self-supervised masked modeling on 367 hours of Utah array data (98 hours of human speech/handwriting recordings and 269 hours of monkey motor task recordings) to enable cross-task (attempted and imagined speech) and cross-species representation transfer; BIT supports two decoding configurations: a cascaded setting (pairing the pretrained encoder with an n-gram LM, achieving state-of-the-art (SOTA) word error rates (WERs) of 5.10% on the Brain-to-Text ’24 holdout set and 2.82% on the ’25 holdout set with model ensembling, outperforming prior methods like Feghhi et al. (2025)) and an end-to-end setting (integrating the encoder with audio large language models (LLMs) such as Aero1-Audio 1.5B, optimized via contrastive learning for cross-modal alignment between neural and text embeddings, which reduces the prior end-to-end WER from 24.69% (Feng et al., 2024) to 10.22% with ensemble, while small-scale audio-LLMs outperform text-based LLMs and BIT aligns neural embeddings of attempted and imagined speech to enable cross-task generalization—especially valuable for low-data imagined speech tasks); the work also notes limitations including slower end-to-end inference (~0.95 seconds per sentence vs. 0.24 seconds for cascaded methods), bidirectional attention unsuitable for real-time decoding, and reliance on limited human data, but overall advances a differentiable, integrable framework for direct neural activity-to-coherent text translation.

**Strengths:**

1. Novel end-to-end BIT framework: Replaces traditional cascaded systems (phoneme decoding followed by n-gram language models) with a single differentiable neural network, solving the dissociation between phoneme error rate (PER) and word error rate (WER) while enabling joint optimization of all stages.
2. Cross-task and cross-species pretrained encoder: Pretrained via self-supervised masked modeling on human (speech-related tasks) and monkey (motor tasks) Utah array data, it outperforms RNNs and Transformers trained from scratch, and sets a new state-of-the-art (SOTA) on the Brain-to-Text benchmarks.
3. Audio LLM integration with contrastive alignment: Integrates audio large language models (LLMs) in the end-to-end setting, and uses contrastive learning to achieve cross-modal alignment between neural and text embeddings; small-scale audio LLMs show better performance in end-to-end decoding than text-based LLMs.
4. Strong cross-task generalization: Aligns the embeddings of attempted and imagined speech (leveraging shared semantic structure), enabling effective decoding of the imagined speech task, which has scarce labeled data.
5. Outstanding performance: In the cascaded setting (especially with model ensembling), it achieves SOTA results on the holdout sets of the Brain-to-Text benchmarks; the end-to-end setting significantly narrows the performance gap with cascaded frameworks.

**Weaknesses:**

1. Slow end-to-end inference: The end-to-end approach is less efficient than the cascaded one, making it unsuitable for real-time brain-computer interfaces (BCIs).
2. Incompatibility with online decoding: The neural encoder uses bidirectional attention to maximize performance, which cannot be applied to online decoding; switching to causal attention will sacrifice some decoding accuracy.
3. Limitations of large models in on-device use: Larger audio large language models (LLMs) cannot run on devices, further restricting real-time applications.
4. Limitations of pretraining data: Human data contribute more to performance gains than cross-species transfer (monkey reaching data are less relevant to speech), and access to private human data for pretraining is limited.
5. Room for improvement in LLM decoders and long-term BCI adaptation: The LLM decoder needs better modality alignment and prompt design; there is also a lack of solutions for non-stationarity, neural plasticity, and user-interface co-adaptation required for long-term BCI use.

**Questions:**

1. Why include monkey motor task data in encoder pretraining? No clear link to human speech neural activity is given.
2. For imagined speech, is "data size difference" controlled when comparing BIT-All and BIT-Cross-Task-Only?
3. Without removing monkey data in ablation, is it proven such cross-species data is essential for pretraining?
4. Why prioritize audio LLMs over text ones? Their bias for neural-to-text decoding lacks clear support.
5. When comparing cascaded/end-to-end frameworks, are LLM differences (size, data) controlled for fairness?

---

> ### Author Response · Authors · 2025-11-20
> **Response to Reviewer GRt5**
>
> We thank the reviewer for highlighting the novelty of our end-to-end framework and our outstanding performance. We address the questions point-by-point below.
>
> > 1. Why include monkey motor task data in encoder pretraining? No clear link to human speech neural activity is given.
>
> Thank you for the question. While this motivation is discussed in the “Related Work” section, we will clarify it in the introduction. Briefly, we include monkey motor-task data in pretraining for two reasons:
>
> (1) **Stable neural representation**: Human intracortical datasets are small and heterogeneous. Adding Utah-array recordings from monkeys (even motor tasks) helps the encoder learn noise-robust and probe-specific spiking patterns to overcome sensor variability. Since SSL does not rely on labels, it focuses on the intrinsic structure of spiking activity rather than spike–to-speech mappings. Such pretraining is valuable given the limited human data.
>
> (2) **Stronger encoder for multimodal alignment**: As shown in prior VLM work [1], encoders pretrained on large, diverse datasets align more easily with LLMs during downstream fine-tuning. Similarly, scaling neural pretraining improves representation quality and facilitates alignment with the audio-LLM decoder.
>
> In addition, prior intracortical BCI studies (POSSAM [2], NDT3 [3]) also used monkey data to mitigate small-data issues, but POSSAM did not transfer to speech decoding, and NDT3 achieved limited cross-species generalization. In contrast, **we demonstrate that large-scale pretraining across species and tasks yields clear transfer benefits, achieving state-of-the-art human speech decoding and ranking #1 on two well-recognized and highly competitive brain-to-text benchmarks.**
>
> > 2. For imagined speech, is "data size difference" controlled when comparing BIT-All and BIT-Cross-Task-Only?
>
> The data size difference between BIT-All and BIT-Cross-Task-Only was not controlled. As noted in the paper, SSL pretraining benefits from using more data to learn stable neural representations, and our goal was to show that including more data beyond same-participant data is helpful. To address your concern, we also ran an experiment comparing SSL and supervised pretraining with equal data sizes. When controlling for data size, we find no substantial performance difference between SL and SSL pretraining for imagined speech decoding. Details can be found in Appendix Section P and Table 9.
>
> > 3. Without removing monkey data in ablation, is it proven such cross-species data is essential for pretraining?
>
> This ablation is already shown in our main results (Figure 2), comparing a neural encoder pretrained only on human data (BIT-Human) with one pretrained on both human and monkey data (BIT-All). Cross-species pretraining yields a small but consistent improvement for attempted speech and a larger gain for imagined speech. This suggests that including cross-species data from the same probe (Utah array) benefits pretraining.
>
> > 4. Why prioritize audio LLMs over text ones? Their bias for neural-to-text decoding lacks clear support.
>
> Thank you for the question. The rationale is explained in the paper (L191–196; L366–370) and summarized here. We prioritize audio LLMs over text-only LLMs because:
>
> (1) Although this is a brain-to-text task, the participant performed a speech task. Neural activity was recorded from motor and speech areas that encode phoneme-level structure [4]. Audio-LLMs, pretrained on speech recognition tasks, naturally capture phonetic and prosodic patterns [5].
>
> (2) Neural signals, like speech waveforms, are continuous time-series data. Speech models thus transfer better to neural activity, whereas text-only LLMs lack the inductive biases needed for temporal signal modeling [6].
>
> > 5. When comparing cascaded/end-to-end frameworks, are LLM differences (size, data) controlled for fairness?
>
> The cascaded models do not use any LLMs, and rely instead on the same 5-gram LM to transcribe phonemes into sentences. For the end-to-end models, we control for LLM differences by using the same audio-LLM decoder, Aero1.5-audio (1.5B parameters), across all results in Figure 2. This ensures performance differences come from the neural encoder, not the LLM. We will clarify this in the main text.
>
> [1] Liu, Haotian, et al. "Visual instruction tuning." NeurIPS 2023.
>
> [2] Hee-Woon Ryoo, Avery, et al. "Generalizable, real-time neural decoding with hybrid state-space models. NeurIPS 2025.
>
> [3] Ye, Joel, et al. "A Generalist Intracortical Motor Decoder." bioRxiv 2025.
>
> [4] Willett, Francis R., et al. "A high-performance speech neuroprosthesis." Nature 2023.
>
> [5] Chu, Yunfei, et al. "Qwen2-audio technical report." arXiv preprint 2024.
>
> [6] Millet, Juliette, et al. "Toward a realistic model of speech processing in the brain with self-supervised learning." NeurIPS 2022.

---

### Author Response · Authors · 2025-11-21
**Response to All Reviewers**

Thank you for your feedback. We have started addressing each reviewer’s comments below. For minor points, such as clarifications, we acknowledge them and will incorporate the changes into the final manuscript. For more in-depth comments, we have added new experiments and analyses to the appendix of the revised manuscript and provided detailed responses when addressing individual reviewer comments. Please let us know if we haven’t fully addressed your questions.

---

### Meta-Review · Area_Chair_tWnk · 2026-01-05

**Summary:**

1. [GRt5, 6BdX, pds5] Concerned about the cross-species and cross-region pretraining lacking justification. Might not data augmentation be sufficient?
2. [6BdX] The motivation for removing time masking during fine-tuning is unclear.
3. [pds5,6BdX] In this limited data application, a fully end-to-end architecture may not be the best choice because such models require more data to train than cascaded models do.
4. [GRt5] For imagined speech, is "data size difference" controlled when comparing BIT-All and BIT-Cross-Task-Only?
5. [GRt5] Why prioritize audio LLMs over text ones? Their bias for neural-to-text decoding lacks clear support.
6. [GRt5] When comparing cascaded/end-to-end frameworks, are LLM differences (size, data) controlled for fairness?
7. [pds5] The paper should employ semantic similarity in addition to word error rate.
8. [pds5] How does performance change as the weighting of the cross-entropy and contrastive losses is changed?
9. [pds5] Training involves hundreds of epochs, raising concerns about overfitting.
10. [pds5] Why use an audio-LLM and not a VLM or Omni model?
11. [pds5] Why use LoRA instead of full fine tuning?
12. [1Ni1] Potential Data Leakage in Pre-training: the SSL model was pre-trained on a large dataset that includes data from some subjects who also appear in the test set.
13. [1Ni1] Characterization of Model Novelty: The Introduction suggests that prior work "does not explore modern architectures, such as Transformer." This statement may be somewhat overstated. The Transformer architecture has been extensively studied across many fields, including neural data modeling.
14. [1Ni1] Limited Theoretical Innovation: The paper primarily combines existing components—SSL pre-training, Transformer encoders, and standard decoding techniques—on publicly available data.
15. [6BdX] Using audio models for supervision is not novel [3], where the results indicate that audio latent guidance is less effective than sequential phoneme labels under phoneme error rate (PER) evaluation.
16. [6BdX] The use of Masked Autoencoder (MAE) pre-training is a well-established technique and does not represent a significant methodological novelty [5,6].
17. [6BdX] The proposed BIT model appears to have a significantly higher training cost than the Time-Masked Transformer baseline. To better contextualize this, the authors should include a runtime comparison, detailing the reported 260-hour pre-training time and the computational overhead of the Audio-LLM Decoder.

**Reviewer Concerns:**

1. This concern is fully addressed in the rebuttal and revision. The authors emphasize that the purpose of the cross-species/cross-region data from monkeys is intended to provide more data representing the properties of the Utah arrays used for recordings, and that the SSL task is appropriate for learning such structure. In addition, the authors argue that training the encoder on more diverse data is expected to simplify the process of aligning the encoder to an LLM, and cite prior work on VLMs as support. They emphasize that ablations show the value of the cross-species data, add scaling curves to the revision to illustrate the influence of the cross-species data on the performance of both end-to-end and cascaded models, and point to state-of-the-art performance on two benchmark tasks. As for data augmentation, the authors reiterate that they apply extensive augmentation and that pretraining, even with cross-species data, outperforms training from scratch.

In followup discussion, Reviewer 6BdX replied that work like POSSAM and NDT3 had already shown the value of cross-species and cross-subject SSL, limiting the novelty of the paper. The authors reply that POSSAM did not investigate speech decoding and NDT3 achieved only limited cross-species generalization while they achieve SOTA results on two highly competitive benchmarks. The AC finds this argument persuasive.

2. This concern is addressed in the rebuttal. The authors explain that time masking during fine-tuning hurts performance, so they disable it.
3. This concern is addressed in the rebuttal. The authors argue that although end-to-end models currently do not perform as well as cascaded models, it is important to study end-to-end models because (1) the availability of intracortical recording data is expected to grow, and end-to-end models are better positioned to exploit larger datasets; (2) end-to-end models can better exploit anticipated intracortial recordings from regions that encode higher level data such as semantics; (3) the cascaded models benefit from a more sophisticated decoding procedure based on beam search and rescoring, while the end-to-end models use simple greedy decoding; (4) in the case of imagined (as opposed to attempted) speech, the performance of end-to-end and cascaded models is comparable; and (5) end-to-end inference is expected to get faster with application of SSM models and GPU-based inference.

In followup discussion, Reviewer 6BdX responded that the current results on the Brain-to-Text benchmarks are insufficient to support the argument that end-to-end models will eventually surpass cascaded models as available datasets grow, and remarked that they understood that a Roformer trained on 595 hours of data from Card et al. achieved a word error rate under 1%. The AC notes that no citation is provided for this <1% WER result. The authors responded that the available public data comprises only 10,948 labeled sentences, so the task remains a small data task. They also responded that their current results clearly show the potential of using more data because their end-to-end results reduce word error rate by more than 50% compared to the prior best end-to-end results.

The AC finds the authors' arguments to be persuasive, especially given the results in many other areas of machine learning (e.g. computer vision, speech recognition, and natural language processing) where end-to-end models have surpassed cascaded models as available training data has grown.

4. This concern is addressed in the rebuttal and revision. The authors say that the point of the comparison between BIT-All and BIT-Cross-Task-Only was to show the value of SSL for using additional data, but they add to the revision a comparison of SL and SSL for imagined speech with comparable amounts of training data, showing that there is no substantial difference in performance in that case.
5. This concern is addressed in the rebuttal and revision. The authors explain that the brain recordings are from motor and speech areas that encode phoneme structure for speech production, so the subjects are performing a speech task for which an audio-LLM may be a better fit. The authors also argue that audio-LLMs have learned to process continuous time-series data, while text LLMs have not, and thus audio-LLMs may be expected to be a better match for the task.
6. This concern is addressed in the rebuttal. The authors point out that cascaded models do not use LLMs, and that in the comparison of different encoders in the end-to-end models, the same audio-LLM is used.
7. This concern is addressed in the rebuttal and revision. The authors argue that semantic similarity measurements are more appropriate for studies involving non-invasive measurements that have lower fidelity, but that with high-fidelity intracortical recordings the word error rates are low enough to be highly meaningful. That said, they add to the revision an analysis of both word and phoneme errors.
8. This concern is addressed in the rebuttal: the authors provide results showing performance for subject T15 as the relative weights on the cross-entropy and InfoNCE losses vary.
9. This concern is addressed in the rebuttal. The authors justify the use of many epochs of training by pointing out that they make heavy use of data augmentation and that they are working with very small training datasets. In addition, model checkpoints for evaluation are selected based on held-out validation set performance to minimize the chances of overfitting.
10. This concern is addressed in the rebuttal and revision. The authors explain that the brain recordings are from motor and speech areas that encode phoneme structure for speech production, so the subjects are performing a speech task for which an audio-LLM may be a better fit.
11. This concern is addressed in the rebuttal. The authors explain that the LLM is too large for full fine-tuning, which would require far more memory and compute (and training data). LoRA enables efficient adaptation with manageable computational cost.
12. This concern is addressed in the rebuttal. The authors point out that in the SSL learning the labels on the speech data are not used, so they are testing transfer from the unlabeled masked prediction task to the labeled neural decoding task. Also, there is so little intracortical recording data available that it is common practice in the field to use data in this manner. Finally, the authors added to the revision some results where the human speech data is excluded from pretraining that show minimal impact.
13. This concern is addressed in the rebuttal: the authors explain that Transformer models are relatively unexplored for processing intracortical recordings.
14. This concern is addressed in the rebuttal and revision. The authors point out that prior studies (POSSAM and NDT3, citations were provided) attempted cross-species transfer to improve human BCIs, but that POSSAM did not transfer to speech decoding and NDT3 achieved limited cross-species generalization. In contrast, the authors point out their work demonstrates that large-scale pretraining across species and tasks provides clear transfer gains, achieving state-of-the-art human speech decoding and ranking #1 on two well-recognized and highly competitive brain-to-text benchmarks. The authors also point out that their study is one of only two that use LLMs to help decode intracortical neural recordings.
15. This concern does not make a lot of sense. The paper cited by the reviewer considers three scenarios: using phonemes, HuBERT discrete units, or discrete articulatory gestures to decode intracortical recordings to text, speech, or animations for a synthetic avatar, respectively. The authors try to address the direct-to-speech aspect, and do so capably, but I am discounting the reviewer's concern.
16. This concern is addressed in the rebuttal. The authors point out that they do not claim the MAE pretraining as a novel contribution and acknowledge that it has been previously applied to intracortical recordings. They clarify that the main contribution of the paper is demonstrating that cross-species and cross-task (and cross-region, as pointed out by the same reviewer) MAE pretraining learns representations of neural activity recorded using Utah arrays that improve decoding of text from such data.
17. This concern is addressed in the rebuttal. First, the authors point out that the reviewer misstated the training cost of the model: the reviewer said 260 hours, but the authors point out it is 48 hours on a single GPU, a figure that is similar to the time-masked Transformer. They then go on to say the goal is to create a foundation model for the BCI community that can be transferred to other datasets with minimal fine-tuning. On decoding overhead, the authors state that inference speed is <1 s for end-to-end versus ~0.25 s for cascaded models and that the end-to-end system can be further accelerated with hybrid architectures (e.g., SSM modules) and modern GPUs, whereas cascaded models remain CPU-bound.

In followup discussion, Reviewer 6BdX clarified that the 260 hours referred to the amount of training data, not to the model training time.

**Reviewer Scores:**

**GRt5** - I am not sure how this reviewer would have responded to the discussion. In my opinion, the authors did an excellent job of addressing this reviewer's concerns, but I also think the reviewer did not have a lot of familiarity with this area.

**pds5** - I believe this reviewer would have increased their score because the authors addressed all of their concerns.

**1Ni1** - I am not sure how this reviewer would have responded to the discussion. In my opinion, the authors did an excellent job of addressing this reviewer's concerns, but I also think the reviewer did not have a lot of familiarity with this area.

**6BdX** - I do not believe that this reviewer would have increased their score given the contentious tone of the discussion with the authors. That said, I think the authors did a good job of addressing the concerns this reviewer raised

---

### Decision · Program_Chairs · 2026-01-26

Accept (Poster)